# OVD-Explorer: A General Information-theoretic Exploration Approach for Reinforcement Learning

## Abstract

Many exploration strategies are built upon the optimism in the face of the uncertainty (OFU) principle for reinforcement learning. However, without considering the aleatoric uncertainty, existing methods may over-explore the state-action pairs with large randomness and hence are non-robust. In this paper, we explicitly capture the aleatoric uncertainty from a distributional perspective and propose an information-theoretic exploration method named Optimistic Value Distribution Explorer (OVD-Explorer). OVD-Explorer follows the OFU principle, but more importantly, it avoids exploring the areas with high aleatoric uncertainty through maximizing the mutual information between policy and the upper bounds of policy's returns. Furthermore, to make OVD-Explorer tractable for continuous RL, we derive a closed form solution, and integrate it with SAC, which, to our knowledge, for the first time alleviates the negative impact on exploration caused by aleatoric uncertainty for continuous RL. Empirical evaluations on the commonly used Mujoco benchmark and a novel GridChaos task demonstrate that OVD-Explorer can alleviate over-exploration and outperform state-of-the-art methods.

## 1 Introduction

The exploration and exploitation trade-off is critical in reinforcement learning (Thompson, 1933; Sutton & Barto, 2018). Many exploration strategies have been proposed in the literature (Osband et al., 2016; Lillicrap et al., 2016; Martin et al., 2017; Pathak et al., 2017; Ciosek et al., 2019). Among these strategies, those following the *Optimism in the Face of Uncertainty* (OFU) principle (Auer et al., 2002) provide efficient guidance for exploration (Chen et al., 2017; Ciosek et al., 2019). Generally, OFU-based methods regard the uncertainty as the result of insufficient exploration for the state-action pair, and refer to such uncertainty as *epistemic uncertainty*. Besides, there is another uncertainty called *aleatoric uncertainty*, which captures environmental stochasticity:

- *Epistemic uncertainty* (a.k.a. parametric uncertainty) represents the ambiguity of the model arisen from insufficient knowledge, and is high at those state-action pairs seldom visited (Dearden et al., 1998; Osband et al., 2016; Moerland et al., 2017).

- *Aleatoric uncertainty* (a.k.a. intrinsic uncertainty) is the variation arisen from environment randomness, caused by the stochasticity of policy, reward and/or transition probability, and is characterized by return distribution. (Bellemare et al., 2017; Moerland et al., 2017).

In the heteroscedastic stochastic tasks where the environment randomness of different state-action pairs differs, those OFU-based methods may be inefficient without properly tackling aleatoric uncertainty (Nikolov et al., 2019). As shown in Figure 1, if the epistemic uncertainty estimation is disturbed due to the volatility caused by aleatoric uncertainty, the exploration strategy (i.e., policy A) optimistic about such estimated uncertainty may lead to explore areas with high aleatoric uncertainty. Visiting such areas results in unstable and risky transitions. In the real world, the aleatoric uncertainty can be caused easily, for example, unpredictable wind can shift the trajectory of an robot's action. If such aleatoric uncertainty is not modelled, the RL agent may be trapped because the state transitions in such area can be wrongly considered novel and worth exploring due to the high uncertainty. This issue, to explore overly the state-action pairs visited frequently but with high

aleatoric uncertainty, is referred to as the *over-exploration issue*. Intuitively, such issue could be solved by avoiding exploring optimistically about aleatoric uncertainty (i.e., policy B in Figure 1).

The similar concern has been raised in the discrete RL, where Nikolov et al. (2019) proposes to use the Information-Directed Sampling (IDS) to avoid the over-exploration issue in the environments with heteroscedastic noise. However, IDS needs to calculate information-regret ratio for each action, thus applying IDS for continuous RL with explosive or even infinite action space could be non-trivial and ineffective. Meanwhile, many advanced continuous RL approaches suffer from the over-exploration issue. Soft Actor-Critic (SAC) is a well performed continuous RL algorithm, but does not account for efficient exploration beyond maximizing policy entropy (Haarnoja et al., 2018). OAC improves the exploration following OFU principle, but it ignores to avoid exploration towards higher aleatoric uncertainty (Ciosek et al., 2019). Therefore, an effective exploration method to address the over-exploration issue for continuous RL is of urgent need.

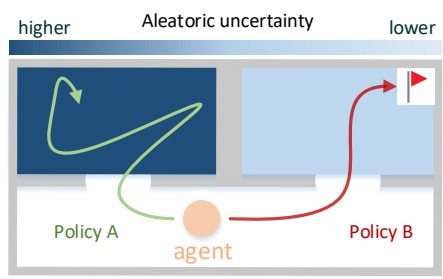

Figure 1: An intuitive example. Red/green lines denote good/bad exploration policies.

To address the over-exploration issue, we propose that the characterization of aleatoric uncertainty and a new exploration principle are necessary. The new principle need to properly trade-off between the epistemic and aleatoric uncertainties, so as to explore seldom visited state-action pairs while avoiding trapped in areas with high aleatoric uncertainty. In this paper, we propose a general information-theoretic exploration approach OVD-Explorer, which enriches the OFU exploration principle with a complement: "avoid areas with high aleatoric uncertainty". Furthermore, to guide the exploration following the new principle, OVD-Explorer maximizes the mutual information between policy and corresponding upper bounds. Notable, the upper bounds represent the best return that policy can reach, formulated using *Optimistic Value Distributions* (OVD), and the value distribution characterises the aleatoric uncertainty. From the theoretical derivation, we show that maximizing such mutual information urges the OVD-Explorer to guide exploration towards the areas with higher epistemic uncertainty and alleviate over-exploration issue by avoiding the areas with lower information gain, i.e., the areas that have higher aleatoric uncertainty but low epistemic uncertainty.

To make OVD-Explorer tractable for continuous action space, we derive its closed form, and propose a scheme to incorporate OVD-Explorer with any policy-based RL algorithm. Practically, we demonstrate the exploration benefits based on SAC. Evaluations on many challenging continuous RL tasks, including a toy task GridChaos, tasks in Mujoco as well as their stochastic variants are conducted, and the results verify our analysis. To the best of our knowledge, OVD-Explorer firstly addresses the negative impact of aleatoric uncertainty for exploration in continuous RL.

## 2 PRELIMINARIES

### 2.1 DISTRIBUTIONAL RL

To model the randomness in the observed long-term return, which characterises aleatoric uncertainty, distributional RL methods (Bellemare et al., 2017; Dabney et al., 2018b;a) are used to estimate $Q$-value distribution rather than expected $Q$-value (Mnih et al., 2015). In our paper, we focus on quantile regression used in QR-DQN (Dabney et al., 2018b), where the distribution of $Q$-value is represented by the quantile random variable $Z$. $Z$ maps the state-action pair to a uniform probability distribution supported on the values at all corresponding quantile fractions. Based on value distribution estimator parameterized as $\theta$, we denote the value at quantile fraction $\tau_i$ as $\hat{Z}_i(s, a, \theta)$ given state-action pair $(s, a)$.

Similar to the Bellman operator in the traditional Q-Learning (Watkins & Dayan, 1992), the distributional Bellman operator (Bellemare et al., 2017) $\mathcal{T}_D^\pi$ under policy $\pi$ is given as:

$$\mathcal{T}_D^\pi Z(s_t, a_t) \overset{D}{=} R(s_t, a_t) + \gamma Z(s_{t+1}, a_{t+1}), a_{t+1} \sim \pi(\cdot|s_{t+1}). \tag{1}$$

Notice that $\mathcal{T}_D^\pi$ operates on random variables, $\stackrel{D}{=}$ denotes that distributions on both sides have equal probability laws. Based on operator $\mathcal{T}_D^\pi$, Dabney et al. (2018b) propose QR-DQN to train quantile estimations via the quantile regression loss (Koenker & Hallock, 2001), which is denoted as:

$$\mathcal{L}_{QR}(\theta) = \frac{1}{N} \sum_{i=1}^{N} \sum_{j=1}^{N} [\rho_{\hat{\tau}_i}(\delta_{i,j})], \tag{2}$$

where TD error $\delta_{i,j} = R(s_t, a_t) + \gamma \hat{Z}_i(s_{t+1}, a_{t+1}; \bar{\theta}) - \hat{Z}_j(s_t, a_t; \theta)$, the quantile Huber loss $\rho_\tau(u) = u * (\tau - \mathbb{1}_{u<0})$, and $\hat{\tau}_i$ means the quantile midpoints, which is defined as $\hat{\tau}_i = \frac{\tau_{i+1} + \tau_i}{2}$.

## 2.2 DISTRIBUTIONAL SOFT ACTOR-CRITIC METHOD

Distributional Soft Actor-Critic (DSAC) (Ma et al., 2020) seamlessly integrates distributional RL with Soft Actor-Critic (SAC) (Haarnoja et al., 2018). Basically, based on the Equation 1, the distributional soft Bellman operator $\mathcal{T}_{DS}^\pi$ is defined considering the maximum entropy RL as follows:

$$\mathcal{T}_{DS}^\pi Z(s_t, a_t) \stackrel{D}{=} R(s_t, a_t) + \gamma[Z(s_{t+1}, a_{t+1}) - \alpha \log \pi(a_{t+1}|s_{t+1})], \tag{3}$$

where $a_{t+1} \sim \pi(\cdot|s_{t+1}), s_{t+1} \sim \mathcal{P}(\cdot|s_t, a_t)$. Then, the quantile regression loss differs from Equation 2 on $\delta_{i,j}$, by extending clipped double Q-Learning (Fujimoto et al., 2018) based on the maximum entropy RL framework, to overcome overestimation of $Q$ value estimation:

$$\delta_{i,j}^k = R(s_t, a_t) + \gamma[\min_{k=1,2} \hat{Z}_i(s_{t+1}, a_{t+1}; \bar{\theta}_k) - \alpha \log \pi(a_{t+1}|s_{t+1}; \bar{\phi})] - \hat{Z}_j(s_t, a_t; \theta_k), \tag{4}$$

where $\bar{\theta}$ and $\bar{\phi}$ represents their target networks respectively. The actor is optimized by minimizing the following actor loss, the same as SAC,

$$\mathcal{J}_\pi(\phi) = \mathbb{E}_{s_t \sim \mathcal{D}, \epsilon_t \sim \mathcal{N}} [\log \pi(f(s_t, \epsilon_t; \phi)|s_t) - Q(s_t, f(s_t, \epsilon_t; \phi); \theta)], \tag{5}$$

where $\mathcal{D}$ is the replay buffer, $f(s, \epsilon; \phi)$ means sampling action with re-parameterized policy and $\epsilon$ is a noise vector sampled from any fixed distribution, like standard spherical Gaussian, and $Q$ value is the minimum value of the expectation on certain distributions, as

$$Q(s_t, a_t; \theta) = \min_{k=1,2} Q(s_t, a_t; \theta_k) = \frac{1}{N} \min_{k=1,2} \sum_{i=0}^{N-1} \hat{Z}_i(s_t, a_t; \theta_k). \tag{6}$$

## 3 METHODOLOGY

We introduce a general exploration method, named OVD-Explorer, to achieve efficient and robust exploration for continuous RL. Overall, OVD-Explorer estimates the upper bounds of the policy's return, and uses the bounds as criteria to find a behavior policy for online exploration from the information-theoretic perspective. Innovatively, to avoid over-exploration, OVD-Explorer particularly takes the aleatoric uncertainty into account, leveraging the optimistic value distribution (OVD) to formulate the upper bounds of policy's return. Then, OVD-Explorer derives the exploration policy via maximizing the mutual information between policy and corresponding upper bounds for an optimistic exploration. Meanwhile, the derived behavior policy avoids being optimistic to aleatoric uncertainty, thus avoiding over-exploration issue.

In the following, we first give the intuition together with the theoretical derivation of OVD-Explorer. Then we practically describe how we formulate OVD based on uncertainty estimation. Lastly, analysis are given to further illustrate why OVD-Explorer can explore effectively and robustly.

### 3.1 OPTIMISTIC VALUE DISTRIBUTION GUIDED EXPLORATION

Most OFU-based algorithms follow the exploration principle: "select the action that leads to areas with high uncertainty". However, this principle is incomplete as the high uncertainty may also be caused by the aleatoric uncertainty, which misleads the exploration direction. Hence, we complement the principle by introducing a constrain: ***"Not only select the action that leads to areas with***

***high uncertainty, but also avoid the ones that only lead to the area with high aleatoric uncertainty"***. Notable, this new principle intuitively describes a good exploration ability, and we will illustrate how to quantitatively measure a policy's exploration ability in the later section. By following this principle, OVD-Explorer derives the behavior policy for exploration.

Intuitively, each policy $\pi$ in the policy space $\Pi$ has different exploration ability (at state $s$), denoted by $\mathbf{F}^\pi(\mathbf{s})$. Given current state $s$, OVD-Explorer aims at finding the exploration policy $\pi_E$ with the best exploration ability by solving the following optimization problem:

$$\pi_E = \arg\max_{\pi \in \Pi} \mathbf{F}^\pi(s). \tag{7}$$

We propose to quantitatively measure the policy's exploration ability from an information-theoretic perspective by measuring the mutual-information of multi-variables as follows:

$$\mathbf{F}^\pi(s) = \mathbf{MI}(\bar{Z}^\pi(s, a_0), \cdots, \bar{Z}^\pi(s, a_{k-1}); \pi(\cdot|s)|s), \tag{8}$$

where $\pi(\cdot|s)$ is the action random variable, and $\bar{Z}^\pi(s, a_i)$ the random variable describing the upper bounds of return that action $a_i$ can reach under policy $\pi$ at state $s$. Here, $a_i \in \mathbf{A}$ denotes any legal action, thus $k$ could be infinite in continuous action space. The higher $\mathbf{F}^\pi(\mathbf{s})$ is, the better ability that policy $\pi$ has in exploring towards higher epistemic uncertainty while simultaneously avoiding higher aleatoric uncertainty, which satisfies the principle we raised. (see analysis in Section 3.3.)

Now we state the Theorem 1 to measure the mutual information above in continuous action space.

**Theorem 1.** *The mutual information in Equation 8 at state $s$ can be approximated as:*

$$\mathbf{F}^\pi(s) \approx \frac{1}{C} \mathop{\mathbb{E}}_{\substack{a \sim \pi(\cdot|s) \\ \bar{z}(s,a) \sim \bar{Z}^\pi(s,a)}} \left[ \Phi_{Z^\pi}(\bar{z}(s, a)) \log \frac{\Phi_{Z^\pi}(\bar{z}(s, a))}{C} \right]. \tag{9}$$

$\Phi_x(\cdot)$ *is the cumulative distribution function (CDF) of random variable $x$, $\bar{z}(s, a)$ is the sampled upper bound of return from its distribution $\bar{Z}^\pi(s, a)$ following policy $\pi$, $Z^\pi$ describes the current return distribution of the policy $\pi$, and $C$ is a constant (See proof in Appendix B.1).*

Theorem 1 reveals that $\mathbf{F}^\pi(s)$ is only proportional to $\Phi_{Z^\pi}(\bar{z}(s, a))$, by maximizing which the policy's exploration ability can be improved. Concretely, given any policy $\pi_\theta$ (parameterized by $\theta$), the derivative $\nabla_\theta \Phi_{Z^\pi_\theta}(\bar{z}(s, a))$ can be measured, thus gradient ascent can be iteratively performed to derive a better exploration policy. (The complete closed form solution is given in later Section 4)

Note that, to perform above optimization procedure, we need to formulate two critical components in $\Phi_{Z^\pi}(\bar{z}(s, a))$: ❶ the return of the policy $Z^\pi(s, a)$ and ❷ upper bound of the return of the policy $\bar{Z}^\pi(s, a)$. Therefore, we first details the formulations in Section 3.2, and then analyze why maximizing $\Phi_{Z^\pi}(\bar{z}(s, a))$ can derive a better exploration in Section 3.3.

## 3.2 FORMULATING THE RETURN OF POLICY AND CORRESPONDING UPPER BOUND

Now, we introduce the formulation of the return of policy $Z^\pi(s, a)$ and corresponding upper bound $\bar{Z}^\pi(s, a)$. As mentioned before, most OFU-based methods neglect the aleatoric uncertainty when formulating the return and upper bound of return (Mavrin et al., 2019; Chen et al., 2017), resulting in the over-exploration issue. Therefore, OVD-Explorer particularly takes into account the aleatoric uncertainty, leveraging distributional RL paradigm to characterize aleatoric uncertainty (Bellemare et al., 2017; Dabney et al., 2018b;a). OVD-Explorer uses two value distribution estimators $\hat{Z}(s, a; \theta_1)$ and $\hat{Z}(s, a; \theta_2)$ parameterized by $\theta_1$ and $\theta_2$, as ensemble estimators to formulate $\bar{Z}^\pi$ and $Z^\pi$ in different ways. Unless stated otherwise, the $(s, a)$ is omitted hereafter to ease notation.

**Formulation of $\bar{Z}^\pi$.** The $\bar{Z}^\pi$ denotes the upper bounds of value that policy $\pi$ can reach via different actions. We propose Gaussian distribution with optimistic mean value for formulation as follows, and accordingly refer to it as *Optimistic Value Distribution* (OVD):

$$\bar{Z}^\pi(s, a) \sim \mathcal{N}(\mu_{\bar{Z}}(s, a), \sigma^2_{\text{aleatoric}}(s, a)), \tag{10}$$

where $\mu_{\bar{Z}}(s, a)$ and $\sigma^2_{\text{aleatoric}}(s, a)$ is mean value and variance, respectively. Notable, Chen et al. (2017) discovers the optimisticity is beneficial for better estimating the upper bound, which motivates us to optimistically estimate the $\mu_{\bar{Z}}(s, a)$ by considering epistemic uncertainty as follows:

$$\mu_{\bar{Z}}(s, a) = \mu(s, a) + \beta\sigma_{\text{epistemic}}(s, a), \text{ s.t. } \begin{cases} \mu(s, a) = \mathbb{E}_{i \sim \mathcal{U}(1,N)} \mathbb{E}_{k=1,2} \hat{Z}_i(s, a; \theta_k) \\ \sigma^2_{\text{epistemic}}(s, a) = \mathbb{E}_{i \sim \mathcal{U}(1,N)} \text{var}_{k=1,2} \hat{Z}_i(s, a; \theta_k) \end{cases}, \tag{11}$$

Table 1: The comparison about two scenarios. Note that for the ease of clarity, without causing ambiguity, we omit part of the notation in the table.

| Fig. 2 | Aleatoric uncertainty $\sigma_a$ | Epistemic uncertainty $\sigma_e$ | Optimistic value estimation $\mu_{\bar{Z}}(s,a)$ | CDF value $\Phi(a)$ | Action $a$ |
|---|---|---|---|---|---|
| (a) | $\sigma_a(a_1) = \sigma_a(a_2)$ | $\sigma_e(a_1) > \sigma_e(a_2)$ | $\mu_{\bar{Z}}(s, a_1) > \mu_{\bar{Z}}(s, a_2)$ | $\Phi(a_1) > \Phi(a_2)$ | $a_1$ |
| (b) | $\sigma_a(a_1) < \sigma_a(a_2)$ | $\sigma_e(a_1) = \sigma_e(a_2)$ | $\mu_{\bar{Z}}(s, a_1) = \mu_{\bar{Z}}(s, a_2)$ | $\Phi(a_1) > \Phi(a_2)$ | $a_1$ |

where $\mu(s,a)$ represents the $Q$-value estimation, $\sigma_{\text{epistemic}}(s,a)$ is the epistemic uncertainty, $\beta$ is a hyper-parameter controlling the magnitude of epistemic uncertainty, $\mathcal{U}$ is uniform distribution, $N$ is the number of quantiles, and $\hat{Z}_i(s,a;\theta_k)$ is the value of the $i$-th quantile drawn from $\hat{Z}(s,a;\theta_k)$.

Moreover, the aleatoric uncertainty should be considered as it will affect the policy's performance, thus we propose to model it as the variance of $\bar{Z}^\pi$. In practice, the aleatoric uncertainty $\sigma_{\text{aleatoric}}(s,a)$ can be captured by value distribution estimators (Clements et al., 2019) as follows:

$$\sigma^2_{\text{aleatoric}}(s,a) = \text{var}_{i\sim\mathcal{U}(1,N)}\left[\mathbb{E}_{k=1,2}\hat{Z}_i(s,a;\theta_k)\right]. \tag{12}$$

Leveraging optimistic value estimations (via epistemic uncertainty) together with explicitly modeling aleatoric uncertainty, the upper bound of the value $\bar{Z}^\pi$ can be effectively formulated as an optimistic Gaussian distribution. Such an upper bound value estimation provides a useful guidance for OVD-Explorer to derive the behavior policy for online exploration.

**Formulation of $Z^\pi$.** $Z^\pi$ estimates the value obtained following policy $\pi$. Inspired by TD3 (Fujimoto et al., 2018), to alleviate overestimation, we propose to formulate $Z^\pi$ in a pessimistic way. In practice, $Z^\pi$ can be measured in two ways. First, similar to formulating $\bar{Z}^\pi$ in Equation 10, $Z^\pi$ can also be formulated as Gaussian distribution as follows:

$$Z^\pi(s,a) \sim \mathcal{N}(\mu_{Z^\pi}(s,a), \sigma^2_{\text{aleatoric}}(s,a)), \quad s.t. \quad \mu_{Z^\pi}(s,a) = \mu(s,a) - \beta\sigma_{\text{epistemic}}(s,a), \tag{13}$$

where $\mu(s,a)$, $\sigma_{\text{aleatoric}}(s,a)$ and $\sigma_{\text{epistemic}}(s,a)$ are the same defined in Equation 11. Differently, $\sigma_{\text{epistemic}}(s,a)$ is subtracted from $\mu(s,a)$, which reveals the pessimistic value estimation of $Z^\pi$.

Another way is to formulate $Z^\pi$ as multivariate uniform distribution as follows:

$$Z^\pi(s,a) \sim \mathcal{U}\{z_i^\pi(s,a;\theta)\}_{i=1,\dots,n}, \quad s.t. \ z_i^\pi(s,a;\theta) = \min_{k=1,2}\hat{Z}_i(s,a;\theta_k), \tag{14}$$

where each quantile value $z_i^\pi(s,a;\theta)$ equals to the minimum estimated value among ensemble estimators (i.e., $z_i(s,a;\theta_k)$). As such, the estimated value of $Z^\pi$ is relative pessimistic.

Adopting pessimistic value estimation avoids overestimation issue, thereby the value distribution $Z^\pi$ can be more accurately formulated. Additional, Gaussian distribution helps more when the environment randomness follows a unimodal distribution, and multivariate uniform distribution can be more flexible and suitable for scenarios with multi-modal distributions. (see more in Section 5.3.)

### 3.3 ANALYSIS OF OVD-EXPLORER

This section analyzes how OVD-Explorer optimistically explores the informative areas and avoids the over-exploration issue. According to Theorem 1, the exploration policy OVD-Explorer derives can maximize $\mathbf{F}^\pi(s)$, which is proportional to CDF value $\Phi_{Z^\pi}(\bar{z}(s,a))$. In the following, Figure 2(a) and 2(b) illustrate such CDF values for different actions (shaded area), and Table 1 shows how uncertainties affect the exploration.

In these cases, the value distribution $Z^\pi(s,a)$ is specified as a Gaussian distribution (in Equation 13), and the sampled optimistic value $\bar{z}(s,a)$ is specified as the mean of OVD $\mu_{\bar{Z}}(s,a)$ (in Equation 11). Specifically, at state $s$, we assume that the means of $Z^\pi$ at actions $a_1$ and $a_2$ are the same for ease of clarification.

Basically, Figure 2(a) shows OVD-Explorer can achieve more optimistic exploration. We assume the aleatoric uncertainty at $a_1$ and $a_2$ are the same, but epistemic uncertainty is higher at $a_1$, causing $\mu_{\bar{Z}}(s, a_1) > \mu_{\bar{Z}}(s, a_2)$. Thus the CDF value is larger at $a_1$, which means the action with higher epistemic uncertainty is preferred.

More crucially, Figure 2(b) shows that OVD-Explorer could guide to avoid the area with higher aleatoric uncertainty given the equal optimism. We assume that the epistemic uncertainty at $a_1$ and $a_2$ are the same, causing their optimistic value estimation to be equal to $\mu_{\bar{Z}}(s, a)$, but aleatoric uncertainty is lower at $a_1$, i.e., PDF curve of $Z^\pi(s, a_1)$ is "thinner and taller". Thus the CDF value is larger at $a_1$, meaning that the action with lower aleatoric uncertainty is preferred.

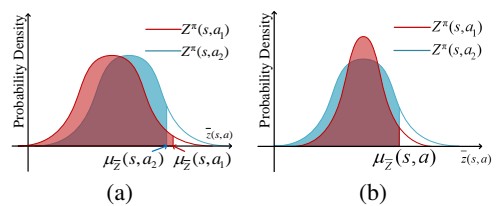

Figure 2: How OVD-Explorer explores (a) optimistically about epistemic uncertainty and (b) pessimistically about aleatoric uncertainty.

In the limit of $t \rightarrow \infty$, the epistemic uncertainty tend to be 0, and the aleatoric uncertainty estimated in Equation 12 converges to represent the true environment randomness. This is comparable to the second case (Figure 2(b)), and OVD-Explorer tends to choose the one with lower aleatoric uncertainty, which is a great advantage of our method over other OFU methods.

To summarize, the exploration guided by OVD-Explorer trades off between two criteria: exploring the areas with higher epistemic uncertainty, to ensure exploring optimistically, and avoiding the areas with higher aleatoric uncertainty, to avoid over-exploration caused by high environment randomness.

## 4 OVD-EXPLORER FOR MODERN RL ALGORITHMS

For continuous action space, the argmax operator in Equation 7 is intractable. To address that, in this section, we derive the behavior policy $\pi_E$ in closed form and state the scheme to incorporate it with existing policy-based algorithms.

First, we denote the policy learned by any policy-based algorithm as the target policy $\pi_T$. To avoid the gap between training data collected using the behavior policy $\pi_E$ and the target policy $\pi_T$, we need to constrain the difference between $\pi_E$ and $\pi_T$, thus we derive $\pi_E$ in the vicinity of $\pi_T$.

Second, to derive exploration $\pi_E$ for modern RL algorithms with stochastic policy based on OVD-Explorer, where both the exploration $\pi_E = \mathcal{N}(\mu_E, \Sigma_E)$ and target policy $\pi_T = \mathcal{N}(\mu_T, \Sigma_T)$ are Gaussian distributions, we introduce the following proposition:

**Proposition 1.** *The OVD-Explorer behavior policy $\pi_E = \mathcal{N}(\mu_E, \Sigma_E)$ is as follows:*

$$\mu_E = \mu_T + \alpha \mathbb{E}_{\bar{Z}^\pi} \left[ m \times \frac{\partial \bar{z}(s, a)}{\partial a} |_{a=\mu_T} \right], \tag{15}$$

*and*

$$\Sigma_E = \Sigma_T. \tag{16}$$

*In specific, $\alpha$ is the step size controlling the exploration level and $m = \log \frac{\Phi_{Z^\pi(s, \mu_T)}(\bar{z}(s, \mu_T))}{C} + 1$ (see proof in Appendix B.2).*

The expectation $\mathbb{E}_{\bar{Z}^\pi}$ can be estimated by sampling K samples, then Equation 15 is simplifies as:

$$\mu_E = \mu_T + \frac{\alpha m}{K} \sum_{i=1}^{K} \frac{\partial \bar{z}_i(s, a)}{\partial a} |_{a=\mu_T}. \tag{17}$$

Algorithm 1 summarizes the overall procedure of OVD-Explorer, including the formulation of $\bar{Z}^\pi$ and $Z^\pi$ (Line 2 and line 3) and behavior policy generation (Line 4). The generated behavior policy can be integrated with any modern policy-based RL algorithms to render the stable and well-performed algorithm. Please see Appendix C for the entire pseudo-code and details about how the OVD-Explorer-based behavior policy is incorporated with DSAC.

## 5 EXPERIMENTS

To reveal the consistency between our theoretical analysis and the performance of OVD-Explorer algorithm, we conduct experiments to address the following questions:

---

**Algorithm 1** The behavior policy (i.e., exploration policy) derived from OVD-Explorer.

---

**Input**: Current state $s_t$, current value distribution estimators $\theta_1, \theta_2$, current policy network $\phi$.
**Output**: Behavior policy $\pi_E$.
 1: Obtain target policy from policy-based RL algorithm $\pi_T(\cdot|s_t; \phi) \sim \mathcal{N}(\mu_T(s_t; \phi), \sigma_T(s_t; \phi))$
 2: Derive OVD $\tilde{Z}^\pi(s_t, \mu_T(s_t; \phi))$ using Eq. 10.
 3: Construct value distribution of behavior policy $Z^\pi(s_t, \mu_T(s_t; \phi))$ using Eq. 13 or 14.
 4: Calculate the mean of behavior policy distribution $\mu_E$ using Eq. 17.
 5: **return** $\pi_E \sim \mathcal{N}(\mu_E, \sigma_T(s_t; \phi))$

---

**RQ1 (Exploration)**: Can OVD-Explorer explore optimistically while avoiding over-exploration simultaneously?
**RQ2 (Performance)**: Can OVD-Explorer handle complex and even stochastic tasks?
**RQ3 (Sensitivity to $\alpha$)**: Is OVD-Explorer sensitive to $\alpha$ that controls exploration magnitude?

## 5.1 EXPERIMENT SETUP

Baselines include SAC (Haarnoja et al., 2018), DSAC (Ma et al., 2020), and DOAC, which is the distributional variant of OAC (Ciosek et al., 2019). We implement Gaussian and quantile formulations of $Z^\pi$ as in Equation 13 and 14, which are denoted in the following as OVDE_G and OVDE_Q, respectively. We test OVD-Explorer on a novel task GridChaos and several tasks in Mujoco (Todorov et al., 2012) including the stochastic variants.

The appendix gives more details, including environment settings, implementation details, hyperparameters settings, and computing infrastructure used, as well as more experiment evaluations, including tasks with different noise scales, tasks with different episode horizon, the sensitivity to $\beta$ and ablation study on the pessimistic formulation of $Z^\pi(s, a)$.

## 5.2 EXPLORATION IN GRIDCHAOS (RQ1)

To illustrate the exploration pattern of OVD-Explorer and show the advantage of OVD-Explorer over DSAC and DOAC, we evaluate OVD-Explorer on a novel continuous and stochastic control task called GridChaos.

Figure 3(a) shows the map of GridChaos, in which we control the cyan triangle (agent) aiming to reach the fixed dark blue goal. The state is the current coordinate, and the action is a two-dimensional vector including the movement angle and distance. One episode terminates when the agent reaches the goal or maximum steps (i.e., MAX_STEP, typically 100). Also, it receives a +100 reward when reaching the goal otherwise 0. The reason why it is *chaos* is that the randomness of the transition is heterogeneous in different parts of this environment. Table 2 shows the environment settings. Moreover, Figure 3(b) shows that OVD-Explorer can reach the goal faster, with more efficient exploration.

Table 2: Settings of GridChaos.

|  | Value | Description |
|---|---|---|
| Observation[0] | [-1, 1] | X-coordinate |
| Observation[1] | [-1, 1] | Y-coordinate |
| Action[0] | [-1, 1] | Degree, mapped to $[-\pi, \pi]$ |
| Action[1] | [-1, 1] | Distance, mapped to [0, MAX_STEP] |
| Noise_0 | 0.5 | Variance of Gaussian noise in the left half of the map (default) |
| Noise_1 | 0.1 | Variance of Gaussian noise in the right half of the map (default) |

Figure 3(c) shows the values of uncertainty estimation and exploration objective (mutual information) taken four different actions at the state shown in Figure 3(a) and at training epoch 1249. Basically, Figure 3(c) illustrates that the estimated aleatoric uncertainty of left is higher than that of right, indicating that OVD-Explorer models aleatoric uncertainty properly. Further, OVD-explorer encourages to explore the right side at that time, since the value of exploration objective (in green) is highest. It implies that OVD-explorer tends to explore areas with higher epistemic uncertainty and avoiding higher aleatoric uncertainty, which is in accordance with our exploration principle. On the other hand, if high epistemic uncertainty is considered only like in OAC(Ciosek et al., 2019), the

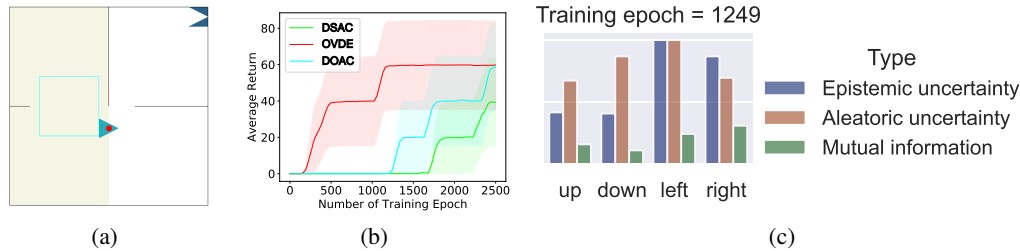

Figure 3: GridChaos. (a) The cyan triangle can move to reach the goal at the top right. (b) The performance on it. (c) The values about uncertainty and exploration objective (mutual information).

agent would be guided to the left. Then the agent may be trapped in the left side due to the high aleatoric uncertainty, and it is the possible reason about why OAC fails tackling such heteroscedastic stochastic task. In appendix, we show more analysis about the exploration pattern, please refer to Appendix E.2 and Appendix E.10.

Besides, we conduct the evaluation when the randomness is high around the goal, as well as many other noise settings, the results enhance the ability of OVD-Explorer. More experiments that evaluate the performance on several other noise settings, as well as more detailed value histogram on different epoch in training process can be found in Appendix.

### 5.3 PERFORMANCE ON MUJOCO TASKS (RQ2)

To demonstrate the performance of OVD-Explorer more generally, we evaluate it on several Mujoco tasks. For 5 standard Mujoco tasks [1], the transition is deterministic and the randomness is only from the stochastic policy. For 5 noisy tasks, the heteroscedastic Gaussian noise is added in Mujoco tasks. Specifically, in each state transition on noisy tasks, Gaussian noise of different scales is randomly injected following a certain probability. Overall, we have the following findings.

Basically, OVD-Explorer can perform stably in the standard tasks with slight or little randomness, guiding efficient exploration. From the results[2] in standard Mujoco tasks in table 3, it is shown that in the relatively easy tasks, i.e., Hopper-v2, Reacher-v2, and HalCheetah-v2, OVD-Explorer does not obtain much gain beyond the baselines, which seems that these tasks are not profoundly demanding for exploration. In the high-dimension tasks, i.e., Ant-v2 as shown in Figure 4(a), OVD-Explorer significantly outperforms DSAC and DOAC. It means that the aleatoric uncertainty caused by policy indeed degrades the performance of DOAC, where the aleatoric and epistemic uncertainties are not distinguished.

Besides, OVD-Explorer can avoid the impact of heteroscedastic aleatoric uncertainty on exploration and thus improve robustness. As the results of noisy Mujoco tasks shown in Table 3, DOAC is worse than DSAC in most cases, which means that heteroscedastic aleatoric uncertainty causes significant degrades of DOAC. Simultaneously, OVD-Explorer significantly outperforms DOAC and DSAC, especially in Noisy Ant-v2 as shown in Figure 4(b).

Furthermore, OVDE_G takes the Gaussian prior and concretely obtains better estimation of critic in the noisy tasks, while OVDE_Q performs better in the standard tasks due to the flexibility of quantile distribution. Generally, the difference between OVDE_G and OVDE_Q is that they use different formulations of $Z^\pi(s, a)$. Theoretically, the value function distribution in OVDE_Q is more flexible and should perform better than OVDE_G, as the reported results on five standard Mujoco tasks show. On the other hand, OVDE_G performs better in the noisy tasks, and it is because that the transition probability is Gaussian in the stochastic Mujoco tasks.

---

[1]https://github.com/openai/gym/tree/master/gym/envs/mujoco

[2]Here, the reported results may be slightly different from previously reported results, partly due to the statistic approach, and partly due to the implementation. Nevertheless, the patterns of these baselines (e.g., DSAC outperforms SAC in the vast majority of cases) are consistent with previous results. The details about the implementation of baselines are described in the Appendix $D.2$.

Table 3: Comparisons of algorithms on five standard and five noisy tasks in Mujoco. We report the averaged performance and standard deviation of 5 runs. Each trail uses the mean undiscounted episodic return over the last 8% epoch (or at most the last 100 epoch) to avoid bias, and the total epoch number is shown in column *epoch*. The maximum value of each row is shown in bold.

| TASK | EPOCH | SAC | DSAC | DOAC | OVD-EXPLORER_G | OVD-EXPLORER_Q |
|---|---|---|---|---|---|---|
| ANT-V2 | 2500 | 4706.2±1338.9 | 6206.9±1202.5 | 6586.7±1023.3 | 7160.6±763.2 | **7590.3**±154.9 |
| HALFCHEETAH-V2 | 2500 | 12373.8±860.9 | 13890.0±3424.4 | 12977.0±140.4 | 14084.5±1579.8 | **14792.4**±997.4 |
| HOPPER-V2 | 1250 | **2751.8**±775.7 | 2199.7±602.7 | 2215.1±557.1 | 2239.5±428.2 | 2619.3±457.0 |
| REACHER-V2 | 250 | -21.6±2.5 | -11.9±0.5 | -19.8±1.7 | -11.6±2.4 | **-10.8**±1.4 |
| INVDBPENDULUM-V2 | 300 | 9344.0±28.4 | **9359.6**±0.1 | 5109.4±3638.7 | 9128.0±460.6 | 9351.3±16.3 |
| N-ANT-V2 | 2500 | 261.5±57.6 | 416.38±42.16 | 337.39±11.96 | **492.54**±50.44 | 450.34±58.42 |
| N-HALFCHEETAH-V2 | 1250 | 351.91±6.68 | 431.39±35.68 | 417.47±39.62 | **445.28**±37.52 | 429.63±34.45 |
| N-HOPPER-V2 | 1250 | 207.06±19.49 | 244.53±4.71 | 242.74±7.87 | **252.09**±7.82 | 237.68±13.11 |
| N-PUSHER-V2 | 1250 | -46.92±12.12 | -25.31±2.46 | -39.13±9.06 | **-23.41**±0.69 | -28.51±4.53 |
| N-INVDBPENDULUM-V2 | 300 | 934.36±1.91 | 932.78±4.02 | 496.61±205.8 | 933.67±1.54 | **934.64**±0.95 |

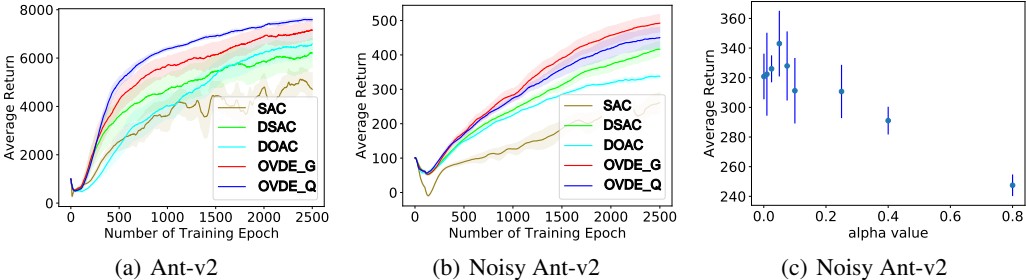

| (a) Ant-v2 | (b) Noisy Ant-v2 | (c) Noisy Ant-v2 |
|---|---|---|

Figure 4: Training curves on (a) Ant-v2 and (b) Noisy Ant-v2. The x-axis indicates the number of training epochs, while the y-axis is the evaluation result represented by the average episode return. The shaded region denotes the half standard deviation of average evaluation over 5 seeds. Curves are smoothed uniformly for visual clarity. (c) Sensitivity to $\alpha$. The x-axis indicates different $\alpha$ settings, while the y-axis is the evaluation result represented by average episode return in the last 100 epoch before total 1250 epoch. Error bars indicate half standard deviation of average evaluation over 5 seeds. The 9 different $\alpha$ values are 0.0005, 0.01. 0.025, 0.05, 0.075, 0.1, 0.25, 0.4, 0.8, respectively.

## 5.4 SENSITIVITY TO $\alpha$ (RQ3)

The step size $\alpha$ in Equation 15 controls how much the behavior policy derived from OVD-Explorer, i.e. $\pi_E$, is far away from the target policy $\pi_T$, which can be essential for exploration benefit. To investigate the appropriate range of it, we test several $\alpha$ value on Noisy Ant-v2 task using OVDE_G, and the result is shown in Figure 4(c). If $\alpha$ is quite small, OVD-Explorer degenerates to DSAC and implies little exploration. In contrast, if $\alpha$ is larger, the performance becomes worse because of the huge gap between behavior policy and target policy. The result demonstrates that $\alpha$ should be taken in a suitable range to facilitate more adequate exploration. In our experiments, we uniformly use $\alpha$ to be equal to 0.05 to show the performance of the OVD-Explorer. In addition, we find that a smaller $\alpha$ leads to higher gains when the task is much more difficult, and the details can be found in Appendix E.6. We also verify the sensitivity of OVD-Explorer to $\beta$ in Appendix E.7 and found that there is a broad range of settings for $\beta$, which can lead to well performance.

## 6 CONCLUSION

This paper proposes an information-theoretic exploration method OVD-Explorer, which introduces a novel measurement of exploration ability, i.e., the mutual information between the policy and the upper bounds of return. By maximizing the mutual information, OVD-Explorer is able to derive the behavior policy, that follows the OFU principle, and further avoids exploring the areas with high aleatoric uncertainty. Integrated with SAC, OVD-Explorer addresses the negative impact of aleatoric uncertainty for exploration in continuous RL for the first time.

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

# A   RELATED WORK

In this work, we consider the exploration strategy under the principle of *Optimism in the Face of Uncertainty* (OFU) (Auer et al., 2002), especially in the heteroscedastic stochastic environment. We aim to improve exploration efficiency, and alleviate the over exploration issue caused by aleatoric uncertainty.

Basic exploration strategies, like $\epsilon$-greedy (Sutton & Barto, 2018), noise perturbation (Lillicrap et al., 2016), entropy regularization (Mnih et al., 2016) and stochastic policy (Haarnoja et al., 2018), lead to undirected exploration through random perturbations. With the increasing emphasis on exploration efficiency in RL, various exploration methods have been developed. One kind of methods uses intrinsic motivation to stimulate agent to explore, such as count-based novelty (Martin et al., 2017; Ostrovski et al., 2017; Bellemare et al., 2016; Tang et al., 2017; Fox et al., 2018), prediction error (Pathak et al., 2017), reachability (Savinov et al., 2019) and information gain on environment dynamics (Houthooft et al., 2016). Some recent methods, originating from tracking uncertainty, guide efficient exploration under the principle of OFU, such as Thompson Sampling (Thompson, 1933; Osband et al., 2016), IDS (Nikolov et al., 2019; Clements et al., 2019) and other customized methods (Moerland et al., 2017; Pathak et al., 2019).

The base of OFU methods is to model epistemic and aleatoric uncertainties in RL. Bootstrapped DQN (Osband et al., 2016) has become the well-used approach for capturing epistemic uncertainty (Kirschner & Krause, 2018; Ciosek et al., 2019), and distributional RL methods (Bellemare et al., 2017; Zhou et al., 2020; Dabney et al., 2018a;b) are used for capturing aleatoric uncertainty. However, most traditional OFU methods do not distinguish the two types of uncertainty, which can easily lead the naive solution to favor actions with higher variances in stochastic tasks, i.e., over-exploration issue.

To address that, Mavrin et al. (2019) study how to take advantage of value distribution for efficient exploration under both types of uncertainty, proposing Decaying Left Truncated Variance (DLTV) based on QR-DQN. Besides, Nikolov et al. (2019) and Clements et al. (2019) propose to use Information Direct Sampling (Kirschner & Krause, 2018) for efficient exploration in RL, which formulates epistemic and heteroscedastic aleatoric uncertainty and maximizes information gain on globally optimal action to explore informative state-action pairs. However, such methods are complicated when deriving a behavior policy and is limited to discrete control.

Meanwhile, there is not any strategy that can help the well-performed continuous RL algorithms (Haarnoja et al., 2018; Ciosek et al., 2019; Ma et al., 2020) to address aleatoric uncertainty when exploration. OAC (Ciosek et al., 2019) proposes exploration bonus guided by the upper bound of $Q$ estimation to facilitate exploration based on Soft Actor-Critic (SAC) (Haarnoja et al., 2018). Nevertheless, OAC ignores the potential impact of the aleatoric uncertainty, which may cause misleading exploration. Our proposed OVD-Explorer is a novel exploration strategy, which can guide agent to explore towards higher epistemic uncertainty, and also avoid the areas with high aleatoric uncertainty, improving the robustness of exploration especially facing heteroscedastic aleatoric uncertainty.

To capture aleatoric uncertainty, OVD-Explorer models the value distribution and uses mutual information to guide exploration following the principle of OFU, measuring the correlations between the policy distribution and upper bounds distribution of return. There are some other information-theoretic exploration strategies using mutual information, such as VIME (Houthooft et al., 2016), which measures the information gain on environment dynamics, and EMI (Kim et al., 2019), which generates intrinsic reward using prediction error of representation learned by mutual information. Those methods can solve sparse reward problem very well by using intrinsic reward. Nevertheless, those exploration methods use mutual information neither on the value distribution, nor for OFU-based exploration. Besides, unlike the mechanisms used in measuring mutual information, such as variational inference (Hinton & van Camp, 1993; Houthooft et al., 2016) and f-divergence (Nowozin et al., 2016; Kim et al., 2019), we find the correlation between policy and upper bounds of return through uncertainty as shown in Theorem 1, thus we can directly derive the close form exploration policy.

# B PROOFS

## B.1 PROOF OF THEOREM 1

In order to prove the Theorem 1, we first propose the following lemma about $\mathbf{F}^\pi(s)$.

**Lemma 1.** *The mutual information of $\bar{Z}^\pi(s, a_0), \cdots, \bar{Z}^\pi(s, a_{k-1})$ and $\pi(\cdot|s)$ at state $s$ is:*

$$\mathbf{F}^\pi(\cdot|s) = \int_{a \sim \pi(\cdot|s)} \mathbb{E}_{\bar{z}(s,a) \sim \bar{Z}^\pi(s,a)} \left[ p(a|\bar{z}(s,a),s) \log \frac{p(a|\bar{z}(s,a),s)}{\pi(a|s)} \right] da, \qquad (18)$$

*where $p(a|\bar{z}(s,a),s)$ represents the posterior probability distribution of policy given current state $s$ and the sampled upper bound of return $\bar{z}(s,a)$.*

*Proof.* For simplicity, we assume that the size of action space is $k = 2$, and the actions are denoted as $a_0$ and $a_1$, $a_0 \neq a_1$. Then we derive the mutual information among three random variables $\mathbf{MI}(\bar{Z}^\pi(s, a_0), \bar{Z}^\pi(s, a_1), \pi(\cdot|s))$, where the action sampled from $\pi(\cdot|s)$ is either $a_0$ or $a_1$.

Considering the formula for the mutual information, $\mathbf{F}^\pi(s)$ is derived as follows:

$$\mathbf{F}^\pi(\cdot|s) = \mathbf{MI}(\bar{Z}^\pi(s, a_0), \bar{Z}^\pi(s, a_1); \pi(\cdot|s)|s)$$

$$= \sum_{\substack{a \sim \pi(\cdot|s) \\ \bar{z}(s,a_0) \sim \bar{Z}^\pi(s,a_0) \\ \bar{z}(s,a_1) \sim \bar{Z}^\pi(s,a_1)}} \left[ p(a, \bar{z}(s, a_0), \bar{z}(s, a_1)) \log \frac{p(a, \bar{z}(s, a_0), \bar{z}(s, a_1))}{\pi(a|s)p(\bar{z}(s, a_0), \bar{z}(s, a_1))} \right]$$

$$= \sum_{\substack{a \sim \pi(\cdot|s) \\ \bar{z}(s,a_0) \sim \bar{Z}^\pi(s,a_0) \\ \bar{z}(s,a_1) \sim \bar{Z}^\pi(s,a_1)}} \left[ p(a|\bar{z}(s, a_0), \bar{z}(s, a_1))p(\bar{z}(s, a_0), \bar{z}(s, a_1)) \log \frac{p(a|\bar{z}(s, a_0), \bar{z}(s, a_1))}{\pi(a|s)} \right]$$

$$= \sum_{a \sim \pi(\cdot|s)} \mathbb{E}_{\substack{\bar{z}(s,a_0) \sim \bar{Z}^\pi(s,a_0) \\ \bar{z}(s,a_1) \sim \bar{Z}^\pi(s,a_1)}} \left[ p(a|\bar{z}(s, a_0), \bar{z}(s, a_1)) \log \frac{p(a|\bar{z}(s, a_0), \bar{z}(s, a_1))}{\pi(a|s)} \right],$$

where the posterior distribution $p(a|\bar{z}(s, a_0), \bar{z}(s, a_1))$ is the probability of choosing action $a$ on the condition of the samples from upper bounds of action $a_0$ and $a_1$.

Considering that in the decision-making process, the probability of action $a_0$ is independent to the upper bound of other actions, such as $\bar{z}(s, a_1)$, which means that $p(a_0|\bar{z}(s, a_0), \bar{z}(s, a_1)) = p(a_0|\bar{z}(s, a_0))$. Therefore, the above equation can be further reduced as follows.

$$\mathbf{F}^\pi(\cdot|s) = \mathbb{E}_{\bar{z}(s,a_0) \sim \bar{Z}^\pi(s,a_0)} \left[ p(a_0|\bar{z}(s, a_0), s) \log \frac{p(a_0|\bar{z}(s, a_0), s)}{\pi(a_0|s)} \right]$$

$$+ \mathbb{E}_{\bar{z}(s,a_1) \sim \bar{Z}^\pi(s,a_1)} \left[ p(a_1|\bar{z}(s, a_1), s) \log \frac{p(a_1|\bar{z}(s, a_1), s)}{\pi(a_1|s)} \right].$$

It is easy to extend to infinite-action case. Based on that for any action $a_i \in [a_0, a_{k-1}]$ and $k \to \infty$, the conditional probability $p(a_i|\bar{z}(s, a_0), \bar{z}(s, a_1), ..., \bar{z}(s, a_{k-1})) = p(a_i|\bar{z}(s, a_i))$, $\mathbf{F}^\pi(\cdot|s)$ can be simplified as following,

$$\mathbf{F}^\pi(\cdot|s) = \int_{a \sim \pi(\cdot|s)} \mathbb{E}_{\bar{z}(s,a) \sim \bar{Z}^\pi(s,a)} \left[ p(a|\bar{z}(s,a),s) \log \frac{p(a|\bar{z}(s,a),s)}{\pi(a|s)} \right] da. \qquad (19)$$

$\square$

Lemma 1 tells that the mutual information $\mathbf{F}^\pi(s_t)$ is in direct proportion to $p(a|\bar{z}(s,a),s)$, which measures how much it is worth acting under the current policy $\pi(a|s)$ when the upper bound is known.

Next, to measure the posterior probability $p(a|\bar{z}(s,a),s)$, we use a general and practically effective approach (Wang & Jegelka, 2017; Belakaria et al., 2020; Perrone et al., 2019; Li et al., 2020) of approximating the posterior probability given upper bound value.

Specifically, we approximate $p(a|\bar{z}(s,a),s)$ using the prior that $z^\pi(s,a) \leq \bar{z}(s,a)$ with given policy $\pi(s,a)$, since $\bar{z}(s,a)$ is the upper bound of $z^\pi(s,a)$. Hence, we use the indicator function $1_{z^\pi(s,a) \leq \bar{z}(s,a)}$ to truncate the policy $\pi(s,a)$, and utilize the constant C to normalize the probability, as is shown in the following equation.

$$p(a|\bar{z}(s,a),s) \approx \frac{1}{C}\pi(a|s)\mathbb{E}_{z^\pi(s,a) \sim Z^\pi(s,a)}\left[1_{z^\pi(s,a) \leq \bar{z}(s,a)}\right].$$

Here, $\mathbb{E}_{z^\pi(s,a) \sim Z^\pi(s,a)}\left[1_{z^\pi(s,a) \leq \bar{z}(s,a)}\right] = \Phi_{Z^\pi(s,a)}(\bar{z}(s,a))$, where $\Phi_x$ is the cumulative distribution function (CDF) of $x$, $\bar{Z}^\pi$ and $Z^\pi$ are the random variables, whose distributions describe the randomness of the returns, and $\bar{z}(s,a)$ is the value of random variable $\bar{Z}^\pi$.

Therefore, the posterior probability can be measured as follows,

$$p(a|\bar{z}(s,a),s) \approx \frac{1}{C}\pi(a|s)\Phi_{Z^\pi}(\bar{z}(s,a)). \tag{20}$$

In our method, we do not use the commonly used mechanisms about mutual information such as neural network estimation (Belghazi et al., 2018) and upper bound estimation (Cheng et al., 2020). Instead, we can find the correlation between random variables as shown in Equation 20, which helps to derive mutual information directly.

According to Lemma 1 and Equation 20, we can give the proof of Theorem 1 in the following.

*Proof.* By Combining Lemma 1 and Equation 20, $\mathbf{F}^\pi(s)$ can be further derived as follows.

$$\begin{aligned}
\mathbf{F}^\pi(s) &= \int_{a \sim \pi(\cdot|s)} \mathbb{E}_{\bar{z}(s,a) \sim \bar{Z}^\pi(s,a)} \left[p(a|\bar{z}(s,a),s)\log\frac{p(a|\bar{z}(s,a),s)}{\pi(a|s)}\right] da \\
&\approx \int_{a \sim \pi(\cdot|s)} \mathbb{E}_{\bar{z}(s,a) \sim \bar{Z}^\pi(s,a)} \left[\frac{1}{C}\pi(a|s)\Phi_{Z^\pi}(\bar{z}(s,a))\log\frac{\pi(a|s)\Phi_{Z^\pi}(\bar{z}(s,a))}{C\pi(a|s)}\right] da \\
&= \int_{a \sim \pi(\cdot|s)} \mathbb{E}_{\bar{z}(s,a) \sim \bar{Z}^\pi(s,a)} \left[\frac{1}{C}\pi(a|s)\Phi_{Z^\pi}(\bar{z}(s,a))\log\frac{\Phi_{Z^\pi}(\bar{z}(s,a))}{C}\right] da \\
&= \frac{1}{C} \mathbb{E}_{\substack{a \sim \pi(\cdot|s) \\ \bar{z}(s,a) \sim \bar{Z}^\pi(s,a)}} \left[\Phi_{Z^\pi}(\bar{z}(s,a))\log\frac{\Phi_{Z^\pi}(\bar{z}(s,a))}{C}\right]
\end{aligned}$$

Here, the last equality follows from Theorem 1. $\qquad\square$

### B.2 PROOF OF PROPOSITION 1

*Proof.* Similar to (Ciosek et al., 2019), we set the covariance matrix of behavior policy $\pi_E$ is that of target policy $\pi_T$, i.e., $\Sigma_E = \Sigma_T$. Hence, the OVD-Explorer problem is simplified as:

$$\begin{aligned}
\mu_E &= \arg\max_\mu \hat{\mathbf{F}}(s) \\
&= \arg\max_\mu, \mathbb{E}_{\bar{Z}^\pi} \left[\Phi_{Z^\pi}(\bar{z}(s,\mu))\log\frac{\Phi_{Z^\pi}(\bar{z}(s,\mu))}{C}\right]
\end{aligned} \tag{21}$$

To ensure that the behavior policy samples actions around the target policy, we derive the $\pi_E$ upon mean $\mu_T$ of target policy $\pi_T$. In specific, we firstly obtain the gradient of $\hat{\mathbf{F}}(s,\mu)$ at $\pi_T$, which is given as follows:

$$\nabla_a \hat{\mathbf{F}}^\pi(s,\mu)|_{\mu=\mu_T} = \mathbb{E}_{\bar{Z}^\pi}\left[\hat{m} \times \frac{\partial \bar{z}(s,a)}{\partial a}|_{a=\mu_T}\right] \tag{22}$$

---

**Algorithm 2** OVD-Explorer for DSAC

---

1: **Initialise:** Value networks $\theta_1, \theta_2$, policy network $\phi$ and their target networks $\bar{\theta}_1, \bar{\theta}_2, \bar{\phi}$, quantiles number N, target smoothing coefficient ($\tau$), discount ($\gamma$), an empty replay pool $\mathcal{D}$
2: **for** each iteration **do**
3:    **for** each environmental step **do**
4:       $a_t \sim \pi_E(a_t, s_t)$ according to Algorithm 1
5:       $\mathcal{D} \leftarrow \mathcal{D} \cup \{(s_t, a_t, r(s_t, a_t), s_{t+1})\}$
6:    **end for**
7:    **for** each training step **do**
8:       **for** i = 1 to N **do**
9:          **for** j = 1 to N **do**
10:             calculate $\delta_{i,j}^k, k = 1, 2$, following Eq. 4
11:          **end for**
12:       **end for**
13:       Calculate $\mathcal{L}_{QR}(\theta_k), k = 1, 2$ using $\delta_{i,j}^k$ following Eq. 2
14:       Update $\theta_k$ with $\nabla\mathcal{L}_{QR}(\theta_k)$
15:       Calculate $\mathcal{J}_\pi(\phi)$, following Eq. 5
16:       Update $\phi$ with $\nabla\mathcal{J}_\pi(\phi)$
17:    **end for**
18:    Update target value network with $\bar{\theta}_k \leftarrow \tau\theta_k + (1 - \tau)\bar{\theta}_k, k = 1, 2$
19:    Update target policy network with $\bar{\phi} \leftarrow \tau\phi + (1 - \tau)\bar{\phi}$
20: **end for**

---

where $\hat{m}$ is given as:

$$\hat{m} = \phi_{Z^\pi(s,\mu_T)}(\bar{z}(s, \mu_T))(\log \frac{\Phi_{Z^\pi(s,\mu_T)}(\bar{z}(s, \mu_T))}{C} + 1), \tag{23}$$

and $\phi(x)$ is the probability distribution function (pdf). Hence, the $\mu_E$ is given as follows:

$$\mu_E = \mu_T + \alpha\mathbb{E}_{\bar{Z}^\pi}\left[m \times \frac{\partial\bar{z}(s, a)}{\partial a}\big|_{a=\mu_T}\right], \tag{24}$$

where $\alpha$ is the step size controlling exploration level and $m = \log \frac{\Phi_{Z^\pi(s,\mu_T)}(\bar{z}(s,\mu_T))}{C} + 1$. $\qquad\square$

## C   ALGORITHM 2: OVD-EXPLORER FOR DSAC

In this section, we show the whole algorithm of our implementation of OVD-Explorer based on DSAC in Algorithm 2. All the code can be found in the supplementary material.

## D   MORE DETAILS ABOUT THE EXPERIMENTS

### D.1   GRIDCHAOS

GridChaos is an environment built on OpenAI's Gym toolkit, whose map is shown as in Figure 3(a) and Figure 5. In this section we illustrate more details in addition to Section 5.2.

The movable cyan triangle and the fixed symmetric dark blue goal are two parts split from square, and the goal is to make the triangle embedded in the goal to recover the original square, which is to say that it is an isosceles triangle whose base side is equal to the height. The triangle is always initialised randomly in the cyan rectangle, and the black line in the map represents the wall, where the triangle will be

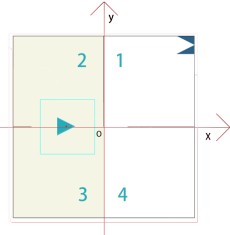

Figure 5: Map of GridChaos.

Table 4: OVD-Explorer parameters

| | Parameter | | Value |
|---|---|---|---|
| Training | Discount | | 0.99 |
| | Target smoothing coefficient | $\tau$ | 5e-3 |
| | Learning rate | | 3e-4 |
| | Optimizer | | Adam (Kingma & Ba, 2015) |
| | Batch size | | 256 |
| | Quantiles amount | | 20 |
| | Replay buffer size | | $1.0 \times 10^6$ for Mujoco tasks |
| | | | $1.0 \times 10^5$ for other tasks |
| | Environment steps per epoch | | $1.0 \times 10^3$ for Mujoco tasks |
| | | | $1.0 \times 10^2$ for other tasks |
| Exploration | Exploration ratio | $\alpha$ | 0.05 |
| | Uncertainty ratio | $\beta$ | 3.2 |
| | Normalization factor | $C$ | 0.5 |

adsorbed once hits the wall. The state transition is stochastic, and we add Gaussian noise to the action resulting in Gaussian transition probability.

To represent the location of the triangle, we establish a Cartesian coordinate system using the centroid of the map as the origin, as shown in Figure 5. Then the coordinates of the triangle are represented by the midpoint of the altitude of the triangle which is shown as the red point in the triangle. In the case shown in Figure 5, the initial coordinate of the triangle (agent) is in the negative half of the x-axis and the task target is in the first quadrant.

## D.2 BASELINES

Ma et al. (2020) shows the performance of DSAC and TD4, which is the distributional extension of TD3 (Fujimoto et al., 2018), and can also be used to capture epistemic and aleatoric uncertainty. Moreover, DSAC outperforms TD4 on Mujoco tasks as shown in Ma et al. (2020), so we evaluate only on SAC and DSAC, and further implement OVD-Explorer based on DSAC to develop the exploration ability.

**SAC.** The SAC (Haarnoja et al., 2018) implementation is mainly based on OAC repository, and the results in Ant-v2 and Hopper-v2 are similar to reported results by OAC. Our SAC report a better result than OAC's implementation for SAC on HalfCheetah-v2, which is because the high variance of this environment as explained as in OAC.

**DSAC.** The DSAC (Ma et al., 2020) implementation is based on our implementation of SAC, except that the distributional Q function in the DSAC repository is used instead of the traditional Q function in SAC. As it is based on SAC, we set the hyper-parameters of DSAC to be consistent with SAC to ensure the fair comparison, which also results in the different reported results from original paper of DSAC. In our results, DSAC can guarantee an absolute advantage over SAC in most cases, which is consistent with the previous conclusion.

**DOAC.** The DOAC implementation is mainly based on our implementation of DSAC as well as the open source code of OAC. As DSAC shows great advantage due to the distributional value estimation, to ensure a fair comparison, we extend OAC (Ciosek et al., 2019) to its distributional version, i.e., DOAC, by replacing the exploration process of DSAC by the behavior policy derived by OAC. We set the hyper-parameters the same as used by OAC in Mujoco[3], and our results of DOAC on Ant-v2 and HalfCheetah-v2 are significantly better than that OAC reported.

---

[3]That is given by the open source code, where $\beta_{\text{UB}}$ is 4.66 and $\delta$ is 23.53

### D.3 IMPLEMENTATION

Our implementation of OVD-Explorer is based on the open source code of OAC [4], also refer to the code of DSAC [5] as well as softlearning [6]. All experiments are performed on NVIDIA GeForce RTX 2080 Ti 11GB graphics card.

The training process of OVD-Explorer and DOAC are the same as in DSAC, except for the different behavior policy used, while OVD-Explorer and DOAC enrich the experience replay with the data using the the derived exploration policies, respectively. To ensure the fair comparison, the hyper-parameters for training process of baselines and OVD-Explorer are the same. Besides, we have three hyper-parameters associated with OVD-Explorer as mentioned before, including $\alpha$ that controls the exploration level, $\beta$ that determines the magnitude of uncertainty we use, as well as $C$ that is the normalization factor. The hyper-parameters in our experiments are shown in Table 4.

## E MORE EXPERIMENT STUDY

### E.1 RUNTIME ANALYSIS

Figure 6 shows the time consumption of algorithms relative to SAC. As can be seen, the distributional value estimation used in DSAC, DOAC and our methods introduces extra time consumption distinctly. Nevertheless, the relative time consumption of OVDE_G and OVDE_Q to SAC is 1.21 and 1.17, respectively, which means that OVD-Explorer spends about 20% more time than SAC to achieve up to nearly 100% performance gain as shown in Figure 6(b). This demonstrates the extra time consumption is well worth it. Besides, the time consumption of OVDE_Q is close to that of DSAC, only with a larger variance, which indicates that the additional time consumption of OVDE_Q is minimal while performing better exploration.

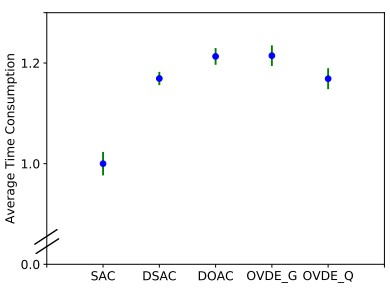

Figure 6: Runtime analysis. The data is from 1 trial of each algorithm on Noisy Ant-v2 task, and the errorbar represents half of the standard deviation.

### E.2 ANALYSIS ABOUT OVD-EXPLORER'S ADVANTAGE IN THE CASE OF GRIDCHAOS

With the heatmap of the visiting frequency of agent during exploration, and the heatmap about the uncertainty estimation, we can visually analyze the patterns and advantages of OVD-Explorer.

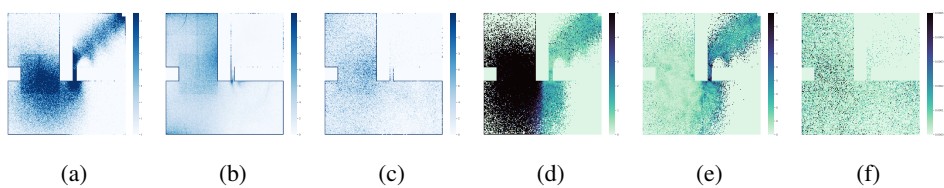

|       |       |       |       |       |       |
|-------|-------|-------|-------|-------|-------|
| (a)   | (b)   | (c)   | (d)   | (e)   | (f)   |

Figure 7: State visiting frequency heatmap from $1.0 \times 10^5$ to $2.5 \times 10^5$ steps of one trial for (a) OVD-Explorer, (b) DSAC and (c) DOAC. (d) Estimated aleatoric uncertainty of OVD-Explorer; (e) Epistemic-aleatoric ratio of OVD-Explorer; (f) Estimated uncertainty for exploration in DOAC.

---

[4]https://github.com/microsoft/oac-explore
[5]https://github.com/xtma/dsac
[6]https://github.com/rail-berkeley/softlearning

The distinctly different exploration patterns can be easily found. Figure 7(a), 7(b) and 7(c) present the state visiting frequency of OVD-Explorer, DSAC and DOAC, respectively. We can see that OVD-Explorer explores directly to the right half, where the environmental randomness is lower, whereas DSAC and DOAC are both stuck in the left half with higher environmental randomness.

Furthermore, we show how OVD-Explorer could explore directly without being trapped by the randomness through the estimated uncertainty. In specific, Figure 7(d) shows that the aleatoric uncertainty estimated by OVD-Explorer is consistent with environment settings, where the environment noise is higher on the left half. Figure 7(e) shows the ratio of estimated epistemic uncertainty and aleatoric uncertainty (i.e., epistemic-aleatoric ratio) of OVD-Explore, and higher ratio means higher epistemic uncertainty or lower aleatoric uncertainty, which is exactly the direction OVD-Explorer explores . The ratio is larger on the right half, which means that OVD-Explorer can avoid being stuck in the left half. Meanwhile, Figure 7(f) presents the estimated uncertainty in DOAC, which is larger on the left half. As DOAC encourages exploring area with relatively large estimated uncertainty, it explains why DOAC is stuck in the left half.

### E.3   EVALUATION ON SEVERAL OTHER NOISE SCALE IN GRIDCHAOS

OVD-Explorer can explore efficiently in heteroscedastic stochastic environment by considering differently about epistemic and aleatoric uncertainty for exploration as shown in Section 5.2. To further empirically prove its strength, we test OVD-Explorer in GridChaos with different noise scales in four quadrants, and the result is shown as Figure 8 and Table 5. We can find that OVD-Explorer can perform well in all those different noise injection of environment.

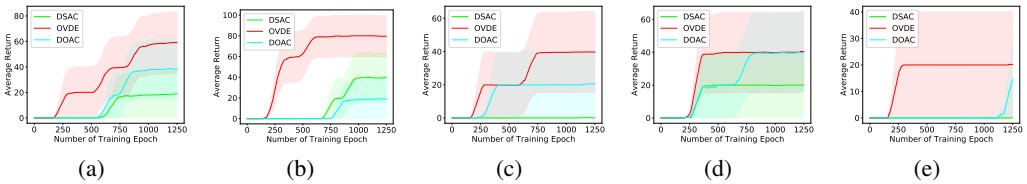

(a) (b) (c) (d) (e)

Figure 8: Training curves on GridChaos with noise of different scale. The x-axis indicates number of training epoch (100 environment steps for each training epoch), while the y-axis is the evaluation result represented by average episode return. The shaded region denotes half standard deviation of average evaluation over 5 seeds. Curves are smoothed uniformly for visual clarity. These results are corresponding to row *A* to row *E* in Table 5.

Table 5: The results for GridChaos. Noise setup shows the different setup for environmental heterogeneous Gaussian noise scale, and the corresponding four columns represent the noise settings in the four quadrants of the Cartesian coordinate system, as shown in Figure 5, in which the goal locates in the first quadrant, and the triangle is initialised in the second or third quadrants. Average return shows the average episodic undiscounted return with half standard derivation in the last 100 epoch before totally 1250 epochs. FRG epoch means the minimum training epoches in the trials used to *Firstly Reach the Goal* before totally 1250 epochs. The row *S* is the standard GridChaos as shown in Figure 3(b), the others are shown in Figure 8

| | NOISE SETUP | | | | AVERAGE RETURN | | | FRG EPOCH | | |
|---|---|---|---|---|---|---|---|---|---|---|
| | 1 | 2 | 3 | 4 | DSAC | OVDE | DOAC | DSAC | OVDE | DOAC |
| S | 0.1 | 0.5 | 0.5 | 0.1 | 0.00±0.00 | **59.30**±48.42 | 3.02±6.03 | 1250+ | **229** | 1222 |
| A | 0.0 | 0.5 | 0.1 | 0.1 | 18.94±37.87 | **58.99**±48.18 | 38.42±47.08 | 1161 | **180** | 662 |
| B | 0.0 | 0.05 | 0.01 | 0.01 | 39.78±48.72 | **79.52**±39.77 | 18.71±37.41 | 694 | **144** | 846 |
| C | 0.05 | 0.1 | 0.1 | 0.05 | 0.05±0.10 | **39.64**±48.55 | 20.59±39.66 | 1250+ | **180** | 309 |
| D | 0.001 | 0.005 | 0.005 | 0.001 | 20.00±40.00 | **40.46**±48.61 | 39.99±48.97 | 284 | **276** | 321 |
| E | 0.0 | 0.0 | 0.0 | 0.0 | 0.00±0.00 | **20.20**±39.90 | 14.60±29.20 | 1250+ | **185** | 1118 |

Concretely, from the results, the following observations deserve to be noticed. First, OVD-Explorer can significantly achieve better average return in all those settings, especially when the noise is set high, and can learn to reach the goal faster (see column about FRG). It shows the ability of OVD-Explorer to guide agent explore against higher aleatoric uncertainty on the left side (the second

and third quadrants). Second, for the task without noise as shown in row *E*, which means the state transition is deterministic, OVD-Explorer still learns quickly. The results in row *E* show the inherently high difficulty of this task, not only because of the very sparse reward, but also the gate leading to the goal is set very small (the width of the gate is only 30% of the length of agent, i.e., the base of the isosceles triangle, which means that at the doorway the agent can only move a very small distance horizontally, otherwise it would be adsorbed to the wall and immobile).

### E.4 EVALUATION IN GRIDCHAOS WHEN THE ALEATORIC UNCERTAINTY IS HIGH AROUND GOAL

In the previous experiments in GridChaos, the noise (i.e., aleatoric uncertainty) near the goal is set lower. In such situation, OVD-Explorer, which follows the principle of OFU and further avoids exploring areas with higher aleatoric uncertainty, could bring significant advantage. Such setup of heterogeneous noise is reasonable, because in real life, the goal or optimal policy is always not expected to be highly stochastic.

Nevertheless, the evaluation about tasks with the existence of high randomness in the target region is valuable, so we conducted the following experiment in GridChaos, where the environment randomness in the right half (first and fourth quadrants), where the target is located, was set larger. The results are shown in Table 6. Note that we use OVDE(P) to denote the usual implementation that pessimistically estimates the value distribution (i.e., using Equation 13). Besides, OVDE(M) denotes the implementation that does not pessimistically estimate the value distribution (i.e., we modify the mean of Gaussian distribution $Z^\pi$ in Equation 13 from the lower bound to expected value of the Q estimation $\mu(s, a)$ as in Equation 11.).

Table 6: The results for GridChaos (additional). This shows row F to J, which are the cases where the optimal policy would face higher aleatoric uncertainty.

| | NOISE SETUP | | AVERAGE RETURN | | | | FRG EPOCH | | | |
|---|---|---|---|---|---|---|---|---|---|---|
| | 1&4 | 2&3 | DSAC | OVDE(P) | OVDE(M) | DOAC | DSAC | (P) | (M) | DOAC |
| F | 0.5 | 0.1 | **50.10**±40.96 | 16.61±33.22 | 17.01±33.86 | **50.52**±41.43 | **479** | 1071 | 583 | **321** |
| G | 0.1 | 0.05 | 0.00±0.00 | 19.84±39.68 | **39.96**±48.95 | 20.14±39.93 | -1 | **188** | 226 | 233 |
| H | 0.05 | 0.005 | 0.00±0.00 | **20.69**±39.61 | 20.07±39.94 | 0.00±0.00 | - 1 | **247** | 308 | -1 |
| I | 0.01 | 0.005 | 0.0±0.0 | 40.00±48.99 | **60.00**±48.99 | 39.99±48.98 | -1 | **200** | 236 | 301 |
| J | 0.005 | 0.001 | 20.00±40.00 | **39.98**±40.97 | 20.00±40.00 | 20.00±40.00 | 236 | 312 | **200** | 296 |

Our experimental findings are mainly the following three aspects.

Firstly, when facing extremely high aleatoric uncertainty around the goal (see row F), which causes the interaction around goal to be very unstable, chaotic and disorder, OVD-Explorer would strongly discourage exploring such a area, and thus performance would be damaged. In contrast, DSAC and DOAC have no restriction on aleatoric uncertainty, and high randomness may instead increase the probability of achieving the goal.

Second, in most cases (see G, H, I, J), OVD-Explorer always can guide better exploration and achieve better performance than DSAC and DOAC, especially when the noise is negligible (see row J). This reflects the fact that our exploration objective (the mutual information shown in Equation 8) makes great sense, achieving an appropriate trade-off between avoiding high aleatoric uncertainty and being optimistic about high epistemic uncertainty.

Third, an interesting finding is that OVD-Explorer may perform better by turning off the pessimistic estimation facing higher aleatoric uncertainty around the goal (see column OVDE(M)). This suggests that excessive pessimism is unnecessary if there is a need to explore areas with high aleatoric uncertainty.

Overall, from the results in Table 5 and Table 6, OVD-Explorer is able to tackle most of the cases quite well. When there is high randomness around the goal, OVD-Explorer has a shortcoming that it will inevitably slow down the efficiency of reaching the goal, because it limit the exploration towards such area. Fortunately, this shortcoming can be mitigated by turning off the pessimistic estimation.

### E.5 EVALUATION OF STATISTICAL SENSE

Table 7: Comparisons of related algorithms on Ant-v2. We report the averaged performance and standard deviation.

| TASK | EPOCH | SEED | DSAC | DOAC | OVD-EXPLORER_G | OVD-EXPLORER_Q |
|------|-------|------|------|------|----------------|----------------|
| ANT-V2 | 2500 | 0, 1, 2, 3, 4 | 6206.9±1202.5 | 6586.7±1023.3 | 7160.6±763.2 | **7590.3**±154.9 |
| ANT-V2 | 2500 | 5, 6, 7, 8, 9 | 6565.0±1343.0 | 6664.2±255.5 | **7190.1**±813.8 | 7174.3±570.0 |
| ANT-V2 | 2500 | 0 - 9 | 6385.9±1287.2 | 6625.4±7446.8 | 7175.3±789.0 | **7382.3**±466.6 |

To counteract the randomness from a statistical perspective, we conduct all experiments for 5 trails with different seeds (typically 0-5), and report the average results with standard deviation. Next, to verify that the 5 trails are sufficient to mitigate the statistical randomness, we run other 5 runs (seeds are set as 5, 6, 7, 8, 9, respectively) for those algorithms on Ant-v2, and show the results in the following. Note that the first row of results is from our previously reported results, which is the same as Table 3, and the second row show the results new. The experimental results in Table 7 show that the results of 5 trials are sufficiently representative of the overall level, while the performance of OVD-Explorer undoubtedly stays ahead.

### E.6 STUDY ON EPISODIC HORIZON

Section 5.3 has shown great advantage of OVD-Explorer over DSAC and DOAC in stochastic Mujoco tasks, which limits the length for an episode to 100 steps. To further empirically verify the efficiency of OVD-Explorer, we test on Noisy Ant-v2 task with different maximum episodic length setup. Our results in Figure 9 show that OVD-Explorer can significantly perform better than baselines in different maximum episodic length (i.e., 250, 500, 750 and 1000). Noting that longer maximum episodic length renders higher difficulty of solving tasks, especially for the high-dimensional tasks demanding exploration. In specific, we have the following two conclusions.

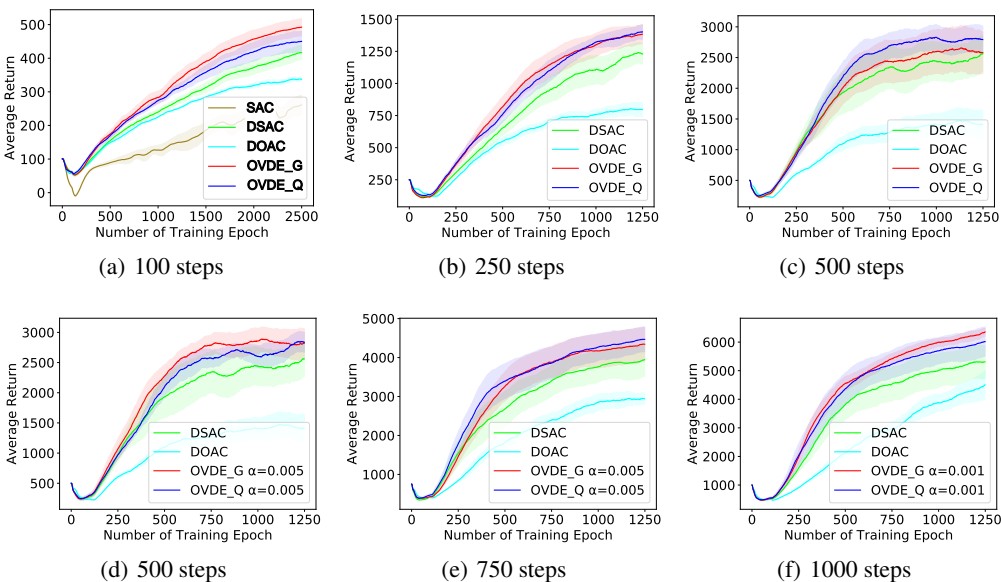

Figure 9: Training curves on Noisy Ant-v2 tasks with different maximum episodic length setup. The x-axis indicates number of training epoch (the number of environment steps for each training epoch is the same as the episodic horizon), while the y-axis is the evaluation result represented by average episode return. The shaded region denotes half standard deviation of average evaluation over 5 seeds. Curves are smoothed uniformly for visual clarity. The sub-title of each figure represents the episodic horizon, also known as the maximum episode length.

Firstly, the exploration should be more conservative in the harder tasks, where we should set smaller $\alpha$ in OVD-Explorer. In Figure 9(a), (b) and (c), $\alpha$ is set to 0.05 by default, while we can find that the advantage of OVD-Explorer_G decreases gradually with the increasing of the difficulty of tasks. Further, if $\alpha$ is set to 0.005, then there is a substantial performance improvement as shown in Figure 9(d). Also, as shown in Figure 9(e), both OVD-Explorer methods perform well when the task episodic horizon is 750 with $\alpha$ set to 0.005. On the hardest task we tested, i.e., the Noisy Ant-v2 with horizon 1000 as shown in Figure 9(f), OVD-Explorer gain remarkable performance, while $\alpha$ is set smaller as 0.001.

Secondly, OVD-Explorer_Q is more stable than OVD-Explorer_G, which is consistent with the conclusion in Section 5.3. We can find from Figure 9(a), (b) and (c) that OVD-Explorer_Q performs stably better while OVD-Explorer_G degrades. OVD-Explorer_G is better in easier task with horizon 100, which is due to the Gaussian prior of noise. But when the task becomes harder, the prior helps less, and OVD-Explorer_Q shows the advantage of more flexibly modeling aleatoric uncertainty and thus the performance is more stable.

### E.7 STUDY ON HYPER-PARAMETERS $\beta$

OVD-Explorer is sensitive to $\alpha$, as shown in Section 5.4. There is another hyper-parameter $\beta$, which controls the scale of uncertainty quantification as shown in Equation 11 and Equation 13, further having an impact on $\bar{Z}^\pi$ and $\underline{Z}^\pi$. To evaluate its sensitivity about $\beta$, we conduct the experiment on Noisy Ant-v2 task using OVD-Explorer_G, and the result is shown in Figure 10. The results demonstrate that there is a broad range of settings for $\beta$, which can lead to well performance.

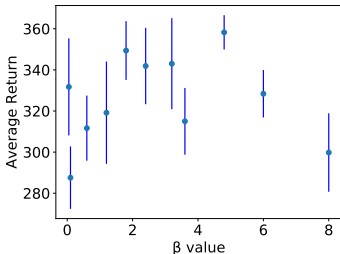

Figure 10: Sensitivity to $\beta$. The x-axis indicates different $\beta$ settings, while the y-axis is the evaluation result represented by average episode return in the last epoch before totally 1250 epoch. Error bars indicate half standard deviation of average evaluation over 5 seeds. The 11 different $\beta$ values are 0.05, 0.1, 0.6, 1.2, 1.8, 2.4, 3.2, 3.6, 4.8, 6.0, 8.0.

### E.8 ABLATION STUDY ON VALUE DISTRIBUTION ESTIMATION

As mentioned in Section 3.2, we estimate $Z^\pi$ pessimistically to alleviate over-estimation. Also, as mentioned in Appendix E.4, the pessimism is unnecessary if there is a need to explore areas with high aleatoric uncertainty. In the following, to investigate the benefit of pessimistic estimation in general case, we compare the performance of OVD-Explorer to the modified versions that use normally estimated $Z^\pi$. Our results show that pessimistic estimation can mostly be better than that using normal estimation.

For OVDE_G (mean), we modified the mean of Gaussian distribution $Z^\pi$ in Equation 13 from the lower bound to average value of the Q estimation $\mu(s, a)$ as in Equation 11. For OVDE_Q (mean), we modified the $z_i^\pi(s, a)$ in Equation 14 from the minimum value to the average value, i.e., $z_i^\pi(s, a; \theta) = \mathbb{E}_{k=1,2}\hat{Z}_i(s, a; \theta_k)$.

As shown in Figure 11, both OVDE_G (mean) and OVDE_Q (mean) perform worse than the pessimistic version. To draw a conclusion, pessimistic estimate is indeed required in general cases. Only when there is a need to explore areas with high aleatoric uncertainty, is such pessimistic estimation worth being turned off.

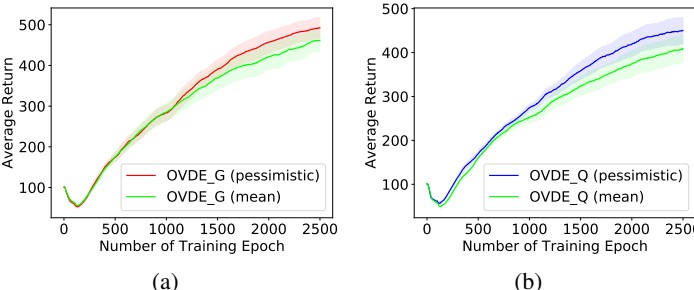

(a)                                      (b)

Figure 11: Training curves on Noisy Ant-v2 with different estimation of $Z^\pi$. The x-axis indicates number of training epoch (100 environment steps for each training epoch), while the y-axis is the evaluation result represented by average episode return. The shaded region denotes half standard deviation of average evaluation over 5 seeds. Curves are smoothed uniformly for visual clarity.

### E.9 THE PERFORMANCE COMPARED WITH RND

RND also follows OFU principle, modeling uncertainty based on network distillation and using it as an intrinsic motivation signal to facilitate agent exploration. For the sake of fairness, we implement RND based on DSAC, denoted by DSAC+RND in the following, and evaluate it on 3 standard Mujoco tasks and 3 noisy Mujoco tasks. We show the results in the following.

Table 8: Comparisons with RND. We report the averaged performance and standard deviation of 5 runs. Each trail uses the mean undiscounted return over the last 100 epoch. The maximum value of each row is shown in bold.

| TASK | EPOCH | DSAC | DSAC+RND | OVD-EXPLORER_G | OVD-EXPLORER_Q |
|---|---|---|---|---|---|
| ANT-V2 | 2500 | 6206.9±1202.5 | 7308.4±641.3 | 7160.6±763.2 | **7590.3**±154.9 |
| HALFCHEETAH-V2 | 2500 | 13890.0±3424.4 | 12198.1±2338.3 | 14084.5±1579.8 | **14792.4**±997.4 |
| HOPPER-V2 | 1250 | 2199.7±602.7 | 2077.9±344.1 | 2239.5±428.2 | **2619.3**±457.0 |
| N-HALFCHEETAH-V2 | 1250 | 431.39±35.68 | 409.48±45.88 | **445.28**±37.52 | 429.63±34.45 |
| N-HOPPER-V2 | 1250 | 244.53±4.71 | 231.46±9.94 | **252.09**±7.82 | 237.68±13.11 |
| N-ANT-V2 (250) | 1250 | 1275.87±172.64 | 1306.05±223.18 | - | **1384.43**±84.39 |

For the complex task Ant-v2, RND brings much greater improvement by facilitating exploration based on the DSAC. For the other simpler tasks, RND does not bring significant performance improvement. This experiment demonstrates to some extent the effectiveness of RND exploration on several Mujoco tasks, but at the same time, our algorithm OVD-Explorer is still better.

### E.10 ANALYSIS ABOUT EXPLORATION PROCESS OF OVD-EXPLORER

In the following, we further verify from statistical analysis that OVD-Explorer is in compliance with our raised exploration principle, i.e., **OVD-Explorer can achieve better trade-off between optimistic exploration and effectively avoiding exploring the areas with high aleatoric uncertainty**. We show the values of uncertainty estimations and our exploration objective (mutual information) at different stages during the training processes of two trials with different noise settings in figures.

In Figure 12, the environment noise is set lower around the goal, noting that the darker background color in the map represents higher aleatoric uncertainty, and the red dot represents the coordinate of the current state. The performance under this trail is given and the agent hardly ever reaches the goal before the 1000th epoch. Therefore, in the early stage, the aleatoric uncertainty is inaccurate and remains very low, as the value distribution shows little divergence. The figure also shows that our exploration objective (in green) is high when the epistemic uncertainty is high. So before the 1000th epoch, the exploration is guided by epistemic uncertainty, which follows the OFU principle. Later, once the goal has been explored, the aleatoric uncertainty is properly modelled, i.e., the aleatoric uncertainty towards left is larger than right at current state (see epoch 1240). Then the mutual information value towards left is lower than right, although the epistemic towards left is higher. It

indicates that OVD-Explorer can property balance epistemic uncertainty and aleatoric uncertainty, and effectively avoid to explore the areas with higher aleatoric uncertainty.

In Figure 13, the environment noise is set higher around the goal. At the stage before the 1000th epoch, the exploration guidance is similar to Figure 12. The agent would hardly estimate the accurate aleatoric uncertainty without obtaining any reward. In the later stage, at the 1240th epoch, OVD-Explorer suggests exploring to the right, even though it has been recognized that the environmental uncertainty on the right is high. This is because the epistemic unertainty dominates under the mutual information. In contrast, at the 1249th epoch, when the action towards right has been explored much, the significant higher aleatoric uncertainty towards right dominates. Therefore, following the mutual information, the action towards left is preferred. This demonstrates the trade off that OVD-Explorer make, which satisfies our raised exploration principle.

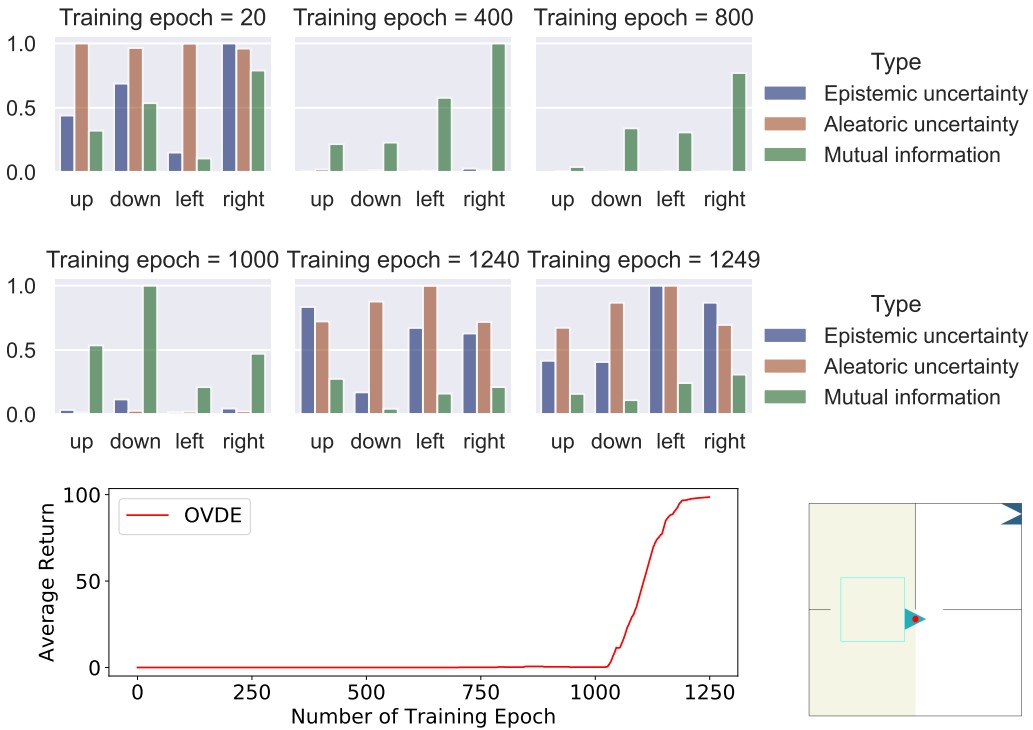

Figure 12: The statistical analysis for the training process, with the aleatoric uncertainty around the goal is set lower.

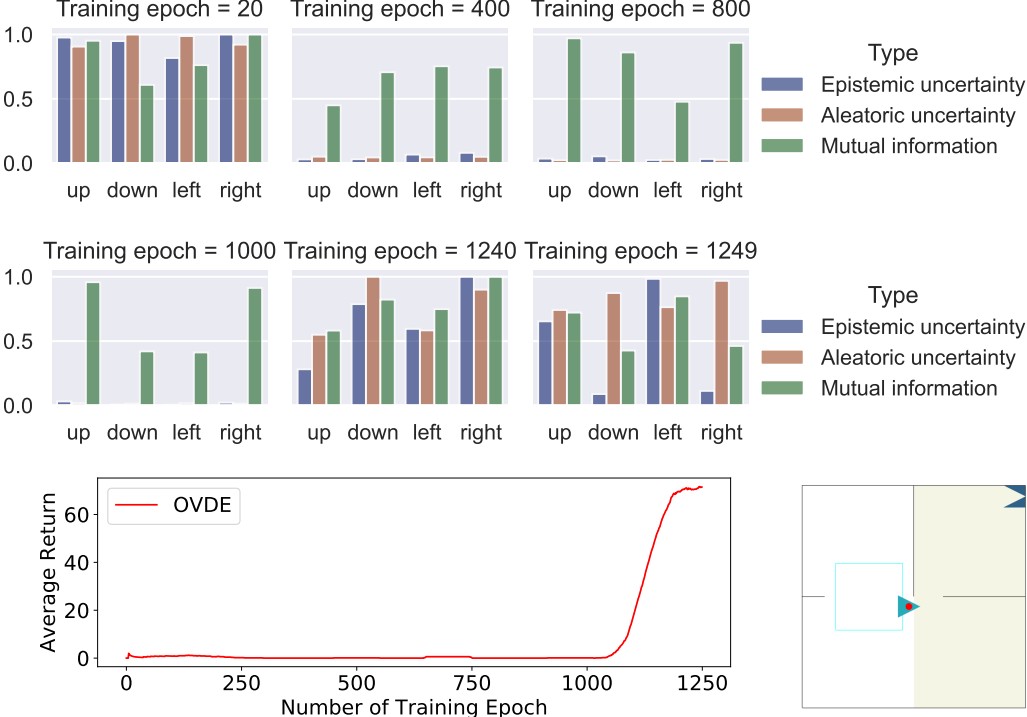

Figure 13: The statistical analysis for the training process, with the aleatoric uncertainty around the goal is set higher.

