# OpenReview forum: "OVD-Explorer: A General Information-theoretic Exploration Approach for Reinforcement Learning"
_ICLR.cc/2022/Conference — ICLR 2022 Submitted_

### Official Review · Reviewer_Sfg5 · 2021-11-02

**Correctness:** 2
**Technical Novelty And Significance:** 3
**Empirical Novelty And Significance:** 3
**Recommendation:** 6
**Confidence:** 3

**Main Review:**

### Strengths
Overall, I appreciate the idea of maximizing mutual information between policy and values and the framework it builds to distinguish epistemic and aleatoric uncertainty, which I consider to be novel. The extensive experiments also show that OVD-Explorer can avoid aleatoric uncertainty while exploring epistemic uncertainty, and has good performance on exploration-demanding environments.

### Weaknesses
I apologize if I made any mistakes here. My major concern is about the technical soundness in paper's theoretical deriviation. In particular, in proof of Lemma 1, it is not clear whether $a$ and $a'$ are fixed or random. In the derivation of $\mathbf{F}^\pi$, it seems $a$ is randomly sampled from $\pi(s)$ while $a'$ is another fixed action, which looks quite wired to me. From my pespective, $a$ and $a'$ should be in a symmetric situation. That is, they should be either both fixed or both random.

Meanwhile, since $p(a\mid \bar{z}(s, a), s)$ is approximated instead of exactly evaluated, it doesn't look appropriate to claim that $p(a\mid \bar{z}(s, a), s)=\frac{1}{C}\pi(a\mid s)\mathbb{E}\_{z^\pi(s, a)\sim Z^\pi(s, a)}\left[\mathbf{1}_{z^\pi(s, a)\leq\bar{z}(s, a)}\right]$.

Furthermore, the writing of this paper may also need some improvements. The detailed suggestions are put in later section.

### Questions
- Connection between law of total variance and two kinds of variances?
- Do $a_0, \dots, a_k$ defined in Eq. (8) enumerate all possible actions in $\mathbf{A}$ or just some subset of $\mathbf{A}$?
- Can OVD-Explorer be applied to value-based algorithm or the case when the action set is discrete and finite? It is not very clear to me how $\frac{\partial \bar{z}(s, a)}{\partial a}$ should be computed under these two scenarios.

### Suggestions on Writing
- Based on the standard notation in information theory, it's probably better to use semicolon ";" to separate $\bar{Z}^\pi(s, a_k)$ and $\pi(s)$ when talking about mutual information.
- Since $\mu_{Z^\pi}$ depends on $\sigma_{\mathrm{epistemic}}$, in Fig. 2(a), assuming means of $Z^\pi$ at $a_1$ and $a_2$ are the same violates Eq. (13). That is, although the conclusion is not affected, PDF curves cannot overlap in Fig. 2(a). It's better to move the curve of $Z^\pi(s, a_1)$ to the left a little bit.
- The statement of Proposition 1 is confusing. Maybe you can simply state that "the gradient of $\hat{\mathbf{F}}^\pi$ with respect to action $a$ at $a=\mu_T$ is ..." in Proposition 1 and then give the formula for $\mu_E$ outside the proposition, which clarifies how you obtain $\mu_E$ by one-step gradient ascent.
- "Target policy" in Algorithm 1 may be confused with target policy network $\bar{\phi}$. Furthermore, it may be better to put Algorithm 2 also into the main context if extra space is allowed later, which can help understand Algorithm 1 a lot.
- It may be better to mention OVDE_G and OVDE_Q in Section 4 and explain that they differ mainly in using Eq. 13 or 14 in line 3 of Algorithm 1.

**Summary Of The Paper:**

This paper proposes a new information-theoretic exploration approach, called OVD-Explorer, for deep reinforcement learning with continuous action space. The OVD-Explorer is capable of exploring epistemic uncertainty while avoiding aleatoric uncertainty. The paper also contains extensive experiments to evaluate the performance of OVD-Explorer.

**Summary Of The Review:**

This paper proposes a novel explorer that can distinguish epistemic and aleatoric uncertainty. Although technically the theoretical motivation does not look very sound to me, the experiment results seem to be quite promising.

---

> ### Author Response · Authors · 2021-11-19
> **Response to Reviewer Sfg5**
>
> We thank the reviewer for the detailed and strong suggestions. we address the questions below.
>
> > **1. About proof section.**
> > **1.1  In proof of Lemma 1, it is not clear whether a and a’ are fixed or random.**
>
> We appreciate your careful review of the theoretical derivation and apologize for the misleading caused by our inadvertent misuse of notation. In fact, $a$ and $a'$ are fixed, and we should not use the same notation with the action sampled from $\pi(\cdot | s)$. We have updated the symbolic representation in the hope that it is no longer ambiguous and misleading. The major updates in the proof section are summarized below:
>
> 1. The notation $a \sim \pi(s)$ or $a \sim \pi(a|s)$ is changed to $a \sim \pi(\cdot | s)$.
> 2. The two actions in Lemma 1 are presented no longer $a$ and $a'$, instead $a_0$ and $a_1$, avoiding ambiguity with the use of same notation for the sample action of $\pi(\cdot|s)$.
> 3. More details are added to show how the two-action case could extend to the infinite-action case.
>
> > **1.2 About $p(a|\bar{z}(s, a), s)$**
>
> It would be more rigorous to use $ \approx$ here, and we have revised it in our paper.
>
> > **2. Connection between law of total variance and two kinds of variances?**
>
> The variance of different $Q$-value estimations, as shown in Eq. 11, represents the differences in the knowledge of different estimators. The more consistent the estimates of different estimators are, the more adequate the knowledge is and the less worth exploring. Such variance is called epistemic uncertainty.
>
> The variance of the individual estimator represents the different $Q$-value estimations for different situations conditioned on the current state-action pair, which is caused by the environment randomness. Such uncertainty is called aleatoric uncertainty, and Eq. 12 measures the average value of aleatoric uncertainty estimated from different estimators.
>
> > **3. Do $a_0, ..., a_k$ defined in Eq. (8) enumerate all possible actions in $\bf{A}$ or just some subset of $\bf{A}$?**
>
> It is all possible actions. As we mentioned in Sec. 3.1 that $a_i \in \bf{A}$ denotes any legal action thus $k$ could be infinite in continuous action space.
>
> > **4. Can OVD-Explorer be applied to value-based algorithm or the case when the action set is discrete and finite?**
>
> OVD-Explorer can surely be applied to discrete control problems. In Sec. 3, we propose that the exploration policy is the one that can maximize such mutual information in Eq. 7, and such argmax operator is naturally applicable for discrete control. Thus, we need to calculate such mutual information (exploration ability) for each action following Theorem 1, and choose the maximum one for exploring the environment.
>
> > **5. Suggestions on Writing**
>
> Thanks for these elaborate suggestions, which we believe can help make the paper clearer and we have addressed them in the revised paper.
>
> We hope the above response has solved your concerns, and we look forward to further discussion with you.

---

> > ### Comment · Reviewer_Sfg5 · 2021-11-30
> > **Response**
> >
> > Thank you very much for your response and my concerns in experiments are well-addressed. However, I strongly suggest to restrict the theory within the case of finite action space and treat the algorithm as a heuristic extension to the continuous action space since the theoretical justification for continuous action space needs a lot more careful treatment. In particular, in the proof of Lemma 1, simply taking $k\rightarrow\infty$ won't give you a result in continuous action space (uncountably infinite) but a result in countably infinite action space; meanwhile, the existence of such limit also needs some argument. Honestly, I haven't seen any work that rigorously studies mutual information about a set of uncountably infinitly many random variables. Could you please give me a reference?
> >
> > However, I'll keep my score unchanged because I think the idea is novel and the experiments look promising.

---

> > > ### Author Response · Authors · 2021-11-30
> > > **Further response to Reviewer Sfg5**
> > >
> > > We sincerely thank you for your further feedback.
> > >
> > > From a purely mathematical point of view, there is hardly any study concerning mutual information about a set of uncountably infinite random variables, because the infinite number of integration operations are unsolvable. Considering the mutual information among infinite random variables $(X_1, X_2, X_3, ...)$, we need to do the integration over the distribution of the $X_1, X_2, X_3, ...$ in turn. Such calculation of the infinite number of integration operations is obviously unsolvable.
> > >
> > > However, in this paper, we were able to measure such a mutual information between infinite random variables specifically because of the fact in RL that the values of other actions $z(s, a_j), j \neq i$ have no effect on the probability of the current action $a_i$, i.e.  $p(a_i|\bar{z}(s, a_0), \bar{z}(s, a_1), ... , \bar{z}(s, a_{k-1})) = p(a_i|\bar{z}(s, a_i))$. As shown in Appendix B.1, such fact greatly simplifies the solution of the particular mutual information we proposed, allowing only the integration over the distribution of the current policy and the values under the particular action sampled from policy to be preserved.

---

### Official Review · Reviewer_xZHr · 2021-11-02

**Correctness:** 2
**Technical Novelty And Significance:** 2
**Empirical Novelty And Significance:** 3
**Recommendation:** 3
**Confidence:** 3

**Main Review:**

The paper considers the important problem of over-exploration, which is perhaps one of the main reasons for the inefficiency of existing online RL algorithms. Good empirical performance is the strong point of this paper.

The approach in this paper is based on the following intuition: "avoid areas with high aleatoric uncertainty". However, I am not sure this intuition is correct.

**Example**
Consider an MDP with actions 0 and 1 from the initial state. Upon taking action 0, the agent moves to a terminal state with a reward of 1. Upon taking action 1, the agent moves to any of the terminal states $s_1$, ..., $s_{10}$ all with probability 1/10. State $s_1$ gives a reward of 100 but states $s_2, ..., s_{10}$  all give zero rewards. In this example, the optimal action is 1 which receives a reward of 10 on average. In this example, the agent must explore the area with high aleatoric uncertainty *even more* to figure out that action 1 can lead to a large reward.

In the paper, the authors write

> This issue, to explore overly the state-action pairs visited frequently but with high aleatoric uncertainty, is referred to as the over exploration issue.

OFU principle constructs confidence bounds based on the samples and high aleatoric uncertainty does not misguide the confidence bounds and result in over exploration. For example, in tabular MDPs, Hoeffding-based exploration, adds a bonus of $\frac{c}{\sqrt{N(s,a)}}$ to the rewards, and the exploration bonus only depends on the number of times $(s,a)$ is visited and it does not depend on aleatoric uncertainty (which is e.g. related to the entropy of $P(s'|s,a)$).

Furthermore, Bernstein-based exploration, adds a bonus of $c \sqrt{\frac{\mathbb{V}_{s'|s,a}(V(s'))}{N(s,a)}}$, which is proved to be information-theoretically optimal up to log factors (see [1]). Notice from this bonus that when the variance of $V(s')$ is larger (which can be attributed to large aleatoric uncertainty), *more exploration is proved to be required*. [2] relates exploration to environmental attributes such as degree of stochasticity.

In OFU, overexploration happens when the constructed upper confidence bound is *loose*, which results in revisiting regions of the MDP that are already explored. This can happen e.g. when coefficient $c$ in the above bonuses is too large.

**Questions/Comments**
- Why is maximizing the multi-variable mutual information in (8) a good idea?
- The exploration strategy has many components such as optimistic estimation of $\mu_{\bar{Z}}$ in (11), pessimistic estimation of $Z^\pi$ in (13), etc. It is unclear which of these components are helping the empirical results.

**References**

[1] Zhang, Zihan, Yuan Zhou, and Xiangyang Ji. "Almost Optimal Model-Free Reinforcement Learning via Reference-Advantage Decomposition." Advances in Neural Information Processing Systems 33 (2020).

[2] Zanette, Andrea, and Emma Brunskill. "Tighter problem-dependent regret bounds in reinforcement learning without domain knowledge using value function bounds." International Conference on Machine Learning. PMLR, 2019.

**Summary Of The Paper:**

The paper is concerned with the problem of over-exploration in RL. The authors propose to capture the aleatoric uncertainty during exploration and propose a new exploration method called Optimistic Value Distribution Explorer (OVD-Explorer). OVD-Explorer is designed to explore optimistically while avoiding the areas with high aleatoric uncertainty. The authors propose a new measure for policy's exploration ability which aims at maximizing the mutual information between policy and policy return upper bounds. OVD-Explorer achieves good empirical performance, outperforming other methods.

**Summary Of The Review:**

The paper presents a new exploration method and shows improved performance compared to state-of-the-art methods. However, the intuition behind the exploration strategy does not seem to be correct.

---

> ### Author Response · Authors · 2021-11-19
> **Response to Reviewer xZHr**
>
> Thanks for your feedback. Below are point-by-point responses to your concerns:
>
> > **1. I am not sure this intuition is correct.**
>
> Firstly, we would clarify that our intuition is to "avoid the actions **only** lead to the areas with high aleatoric uncertainty", which means that OVD-explorer trade-off the uncertainties and avoid exploring the areas with relatively high aleatoric uncertainty and low epistemic uncertainty, i.e., lower information gain. OVD-explorer will not simply "avoid the high aleatoric areas". On the contrary, OVD-Explorer does explore much in the areas with a larger degree of stochasticity, as we analyzed in Appendix E.2, E.5, and E.10.
>
> Besides, for the example you mentioned, OVD-Explorer would not completely avoid exploring action 1 that has high aleatoric uncertainty. The reasons are as follows.
> - At the beginning of exploration, the agent can not be aware that the aleatoric uncertainty is higher in action 1. Thus the exploration is mainly guided towards areas with higher epistemic uncertainty (i.e., being visited less often), then both actions may be explored several times.
> - Besides, as analyzed in Sec. 3.3, OVD-Explorer trades off between two criteria: higher optimistic value sampled from OVD and lower aleatoric uncertainty, meaning that dominated higher optimistic value of action 1 may lead the exploration.
> - Also, we have conducted experiments in a similar situation when the aleatoric uncertainty is high around the goal, and the results show that OVD-Explorer can handle such situations well. Please refer to Appendix E.5 for more details.
>
> Moreover, such intuition is similar to several previous works.
> IDS [1] proposed that these approaches do not appropriately exploit the heteroscedastic nature of the return, and noisier actions are more likely to be chosen, which can slow down learning.
> DLTV [2] evaluated a naive approach that uses the variance of the estimated value distribution as a bonus, but showed its bad performance, which is due to the aleatoric uncertainty.
>
> > **2. About Hoeffding-based exploration and Bernstein-based exploration.**
>
> First, we think the assumptions for Bernstein-based exploration cannot be retained in OVD-Explorer. The regret bound of Bernstein-based exploration is proved on tabular MDP, but our paper mainly focuses on the continuous action and state space. Moreover, the variance of the value function is assumed to be upper bounded by the square of the horizon length, based on the assumption of MDP Regularity [3]. However, it is not the case in our paper, where OVD-Explorer works in the discounted infinite-horizon MDP and the variance of value function could be unbounded, which breaks the assumption in Bernstein-based exploration.
>
> Second, Hoeffding-based exploration is very like OAC, which only considers epistemic uncertainty. And the experiments in Sec. 4 show that OVD-Explorer is better than OAC.
>
> > **3. Why is maximizing the multi-variable mutual information in (8) a good idea?**
>
> We’d like to clarify that mutual information to optimal value function[4] (or action[5]) is widely used to design exploration policy and achieve relatively good performance. The potential for OVD-explorer to tackle with the over-exploration issue is attributed to approximating the optimal value function by formulating epistemic and aleatoric uncertainty explicitly. Further, the detailed analysis in Sec. 3.3 and experiments in Sec. 4 show that OVD-explorer avoids the low information gain areas with relatively high aleatoric uncertainty and low epistemic uncertainty.
>
> > **4. It is unclear which of these components are helping the empirical results.**
>
> Firstly, optimistic estimation shows potential to improve the exploration. As shown in Eq. 11, if the optimistic estimation is not used and the expected value estimation is used instead, OVD-Explorer will degrade to DSAC. The results show that OVD-Explorer outperforms DSAC, implying that optimistic estimation improves the exploration.
>
> Further, the pessimistic estimation is for alleviating overestimation as mentioned in the paper, and we also conducted the ablation studies in Appendix E.4 and E.8. The conclusion is that the pessimistic estimate is indeed required in general cases.
>
> Hope our responses would clarify your concerns. We are looking forward to your future comments.
>
> ---
> [1] Kirschner, Johannes, and Andreas Krause. "Information directed sampling and bandits with heteroscedastic noise." Conference On Learning Theory. PMLR, 2018.
>
> [2] Mavrin, Borislav, et al. "Distributional reinforcement learning for efficient exploration." ICML. PMLR, 2019.
>
> [3] Azar, Mohammad Gheshlaghi, Ian Osband, and Rémi Munos. "Minimax regret bounds for reinforcement learning." ICML. PMLR, 2017.
>
> [4] Wang, Zi, and Stefanie Jegelka. "Max-value entropy search for efficient Bayesian optimization." ICML. PMLR, 2017.
>
> [5] Russo D, Van Roy B. Learning to optimize via information-directed sampling. Advances in NIPS, 2014, 27: 1583-1591.

---

### Official Review · Reviewer_YS7S · 2021-11-02

**Correctness:** 3
**Technical Novelty And Significance:** 3
**Empirical Novelty And Significance:** 3
**Recommendation:** 6
**Confidence:** 3

**Main Review:**

**Pros**

* The proposed OVD-Explorer exploration algorithm is able to force the agent to visit state-action pairs that are not visited frequently, while at the same time avoiding areas with high aleatoric uncertainty. In general the idea of using mutual information is quite novel and interesting.
* A theoretical analysis is also provided verifying that the over-exploration issue can be tackled through the maximisation of such mutual information.
* The empirical results on the noisy variants of the five MuJoCo environments show the efficiency of the OVD-Explorer compared to the other baselines.
* In general the paper is well written. The proposed algorithm and the empirical results are presented in a clear way.

**Cons - Comments**

* The OVD-Explorer exploration algorithm has been tested only on five domains. The evaluation of  OVD-Explorer in more domains is necessary, e.g., DeepMind control suite.
* The authors should make more clear the problem of aleatoric exploration. When do we encounter this kind of problem in the real world? Some real world use cases should be presented making clear the necessity of this kind of exploration algorithmic schemes.
* How easy is for OVD-Explorer to be integrated with any other RL algorithm apart from SAC?
* The reasoning behind the usage of mutual information for alleviating the aleatoric uncertainty is not totally clear.



**Summary Of The Paper:**

This paper introduces OVD-Explorer, an efficient exploration algorithm that is able to detect and avoid areas with high aleatoric uncertainty. Actually, the proposed algorithm explores state-action pairs that have not been visited frequently (high epistemic uncertainty) and at the same time avoids areas with high aleatoric uncertainty. In order to achieve this behavior, OVD-Explorer maximises the mutual information between policy and corresponding upper bounds. Theoretical results show that the over-exploration issue can be tackled through the maximisation of such mutual information. Empirical analysis has been conducted on a toy task and at five MuJoCo environments including their stochastic variants.


**Summary Of The Review:**

In general the contributions of this work are significant and novel. The proposed OVD-Explorer exploration algorithm seems to be efficient on the noisy variants of the five MuJoCo environments, showing that it is able to avoid areas with high aleatoric uncertainty. Apart from that a theoretical analysis is also provided. The main weakness of this work is the fact that the OVD-Explorer has been tested only on five MuJoCo environments. I would have expected the empirical analysis to have been conducted on more environments, e.g., DeepMind control suite. Last but not least, authors should make more clear the problem of aleatoric exploration by providing some real-world cases where we encounter this type of problem.

---

> ### Author Response · Authors · 2021-11-19
> **Response to Reviewer YS7S**
>
> Thanks for your time to read our paper thoroughly. Our responses are as follows.
>
> > **1. The OVD-Explorer exploration algorithm has been tested only on five domains. The evaluation of OVD-Explorer in more domains is necessary, e.g., DeepMind control suite.**
>
> We have conducted various evaluations on GridChaos and Mujoco, from diverse perspectives such as with or without noise, different noise scales, and different horizon scales. The results consistently demonstrate the effectiveness of OVD-Explorer. Besides, following your suggestion, we are conducting experiments in DeepMind control suite. The very beginning results are shown in the following table, noting that the hyper-parameters are consistent with the evaluation in Mujoco, and the form of the table is consistent with Table 3 in our paper. We will keep refining the table below and eventually attach it to the appendix of the paper.
>
> | Task               | Epoch | DSAC          | DOAC          | OVD-Explorer_G | OVD-Explorer_Q |   |   |   |   |
> |--------------------|-------|---------------|---------------|----------------|----------------|---|---|---|---|
> | Finger.spin        | 500   | 929.18±47.12  | 925.38±9.21   | **985.39**±0.78    | 924.35±60.20   |   |   |   |   |
> | N-Finger.spin      | 500   | **983.53**±3.61   | 919.91±3.18   | 951.06±44.50   | **980.98**±6.57    |   |   |   |   |
> | Finger.turn_easy   | 500   | 629.73±97.27  | 616.81±132.71 | 663.00±125.44  | **676.57**±92.16   |   |   |   |   |
> | N-Finger.turn_easy | 500   | **679.39**±166.19 | 637.24±184.65 | 612.24±178.43  | **672.15**±218.38  |   |   |
> | Hopper.stand | 500   | 559.04±304.23 | 507.80±215.92 | 649.94±325.79  | **692.14**±270.66  |   |   |
> | N-Hopper.stand | 500   | 577.76±278.20 | 497.01±213.14 | 513.45±344.09  | **595.80**±197.23  |   |   |
>
> > **2. Some real world use cases should be presented making clear the necessity of this kind of exploration algorithmic schemes.**
>
> We added such a description in the revised version.
>
> In the real world, the aleatoric uncertainty can be caused easily, for example, unpredictable wind can shift the trajectory of the robot's action, and rough ground can change the force point of the object, etc. If such aleatoric uncertainty is not modeled, then the RL agent may be trapped because each state transition in such area can be wrongly considered novel and worth exploring due to the high uncertainty. And our intuition is to make the agent aware of areas with high uncertainty that is mainly caused by aleatoric uncertainty (environment randomness), instead of epistemic uncertainty, which is not worth exploring, thus enhancing exploration efficiency.
>
> > **3. How easy is for OVD-Explorer to be integrated with any other RL algorithm apart from SAC?**
>
> The core of OVD-Explorer being integrated with any other RL algorithm is the exploration bonus (the term weighted by $\alpha$ in Proposition 1). And, based on the formulation of $\bar{Z}(s, a)$ and $Z^\pi(s, a)$, such exploration bonus can be directed calculated.
>
> For example, for integrating TD3 [1] with OVD-Explorer, we need to model the value distribution firstly, and we get the distributional version of TD3 [2]. Then, we need to formulate value distribution $Z^\pi(s, a)$ and OVD $\bar{Z}(s, a)$ following Sec. 3.2 and calculate the **exploration bonus** following Eq. 17. To integrate OVD-Explorer, we only need to replace the noise added on policy, by the **exploration bonus** we derived.
>
> > **4. The reasoning behind the usage of mutual information for alleviating the aleatoric uncertainty is not totally clear.**
>
> We’d like to clarify that mutual information to optimal value function[3] (or action[4]) is widely used to design exploration policy and achieve relatively good performance. The potential for OVD-explorer to tackle with the over-exploration issue is attributed to approximating the optimal value function by formulating epistemic and aleatoric uncertainty explicitly. Further, the detailed analysis in Sec. 3.3 and experiments in Sec. 4 show that OVD-explorer avoids the low information gain areas with relatively high aleatoric uncertainty and low epistemic uncertainty.
>
> We hope that we have addressed your concerns and look forward to discussing in further.
>
> ---
> [1] Scott Fujimoto, Herke van Hoof, and David Meger. Addressing function approximation error in actor-critic methods. In Proceedings of the 35th International Conference on Machine Learning, ICML 2018, Stockholmsmassan, Stockholm, Sweden, July 10-15, 2018.
>
> [2] Xiaoteng Ma, Qiyuan Zhang, Li Xia, Zhengyuan Zhou, Jun Yang, and Qianchuan Zhao. Distributional soft actor critic for risk sensitive learning. CoRR, abs/2004.14547, 2020.
>
> [3] Zi Wang and Stefanie Jegelka. Max-value entropy search for efficient Bayesian optimization. volume 70 of Proceedings of Machine Learning Research, pages 3627–3635, Aug 2017. PMLR
>
> [4] Russo D, Van Roy B. Learning to optimize via information-directed sampling. Advances in NIPS, 2014, 27: 1583-1591.

---

### Official Review · Reviewer_nxh5 · 2021-11-10

**Correctness:** 2
**Technical Novelty And Significance:** 1
**Empirical Novelty And Significance:** 2
**Recommendation:** 3
**Confidence:** 3

**Main Review:**

This paper studies the over-exploration issue in RL, which is important to designing good exploration algorithms. The paper proposes an information-driven exploration algorithm OVD-Explorer that has potential to deal with the over-exploration issue. Here are some of my questions and comments.
1. The RHS of Equation (7) depends on state $s$, so I think the LHS should also depend on $s$. Equation (7) is the key optimization problem that OVD-Explorer needs to solve, and this maximization is over the policy space which could be much more complicated than the action space. Can the authors comment on the computational complexity of OVD-Explorer? In addition, comparing to information-directed sampling (IDS) that optimizes over the action space, does OVD-Explorer have advantages in terms of computational complexity?
2. Besides the version of IDS proposed [1], [2] also proposes the so-called value-IDS for RL. I think it is necessary to compare OVD-Explorer with other information-theoretic algorithms, especially through comprehensive numerical experiments. Indeed, both OVD-Explorer and value-IDS use Gaussian distributions to to approximate the true distributions of state-action values.
3. Honestly speaking, I had a hard time verifying the properties of OVD-Explorer (e.g., Theorem 1, Lemma 1). In Theorem 1, when $k$ is finite, it is fine, but when $k$ is infinite, things need to be written more carefully and rigorously. Also, please elaborate more the constant $C$ and how it is related to other elements. In Lemma 1, does $\sum_{a\sim\pi(a|s)}$ implies this result only holds for finite-action case? Btw, the notation $a\sim\pi(a|s)$ is a bit weird since both sides has $a$ there. In the proof for two-action case (i.e., $k=2$), there are actions $a$ and $a'$, but why the summation is over $a\sim\pi(s)$? Btw, what is the difference between $\pi(s)$ and $\pi(a|s)$? I am confused with the proof for two-action case and how it could be extended to infinite-action case. I hope these proofs could be more readable and verifiable.
4. I think it is necessary to elaborate more on why OVD-Explorer has potential to deal with the over-exploration issue. I could not get from the paper the intuitions why maximizing the mutual information between policy and the upper bounds of policy's returns can make OVD-Explorer avoid exploring the areas with high aleatoric uncertainty.
5. It would be great that some theoretical guarantee (e.g. regret bound, sample complexity) can be showed for OVD-Explorer
6. There is no need to introduce the abbreviation OVD several times in the main text.


[1] Nikolay Nikolov, Johannes Kirschner, Felix Berkenkamp, Andreas Krause, Information-Directed Exploration for Deep Reinforcement Learning, ICLR 2019.
[2] Xiuyuan Lu, Benjamin Van Roy, Vikranth Dwaracherla, Morteza Ibrahimi, Ian Osband, Zheng Wen, Reinforcement Learning, Bit by Bit, 2021.

**Summary Of The Paper:**

The paper proposes the so-called Optimistic Value Distribution Explorer (OVD-Explorer) to deal with over-exploration issue in reinforcement learning (RL). It is claimed that the proposed algorithm is tractable for continuous action space and can avoid exploring  the areas with high aleatoric uncertainty. The paper tests the performances of OVD-Explorer and some of its competitors on some common environments.

**Summary Of The Review:**

This is an interesting paper, but it seems that some improvements are needed.

---

> ### Author Response · Authors · 2021-11-19
> **Response to Reviewer nxh5**
>
> Thank you for reviewing our paper and giving detailed concerns, we reply to each of them as follows.
>
> > **1. I think the LHS should also depend on s. Does OVD-Explorer have advantages in terms of computational complexity over IDS that optimizes over the action space?**
>
> Surely that the LHS of Eq. 7 depends on state s.
> Also, since we mainly focus on the continuous action space, optimizing over policy space is widely used and achieves relatively good performance.
>
> Moreover, to fairly analyze the computational complexity, we compare OVD-Explorer to DSAC in terms of time consumption, and the results are shown in Appendix E.1. OVD-Explorer is close to that of DSAC, only with a larger variance,  which indicates that the additional time consumption of OVD-Explorer is minimal while performing better exploration.
>
> > **2. I think it is necessary to compare OVD-Explorer with other information-theoretic algorithms, especially through comprehensive numerical experiments.**
>
> As you mentioned that [1] and [2] propose methods to use IDS for exploration in different ways. However, IDS methods are mainly for discrete control, instead, OVD-Explorer focuses mainly on continuous control problems.
>
> Moreover, the code for [1] is public but is only for discrete control and the code for [2] has not been released yet. There is no standardized implementation of IDS for the continuous control problem in the RL community that can guarantee performance.
>
> Certainly, if an implementation of IDS on the continuous control problem is released in the RL community, we would be ready to conduct experiment comparisons. By the way, our code has also been publicly available.
>
> > **3. About the proof section.**
>
> We have reviewed the proof very carefully and made the following changes. We hope that our revised version is clear and easy to read.
>
> 1.The notation $a \sim \pi(s)$ or $a \sim \pi(a|s)$ is changed to $a \sim \pi(\cdot | s)$.
>
> 2. The two actions in Lemma 1 are presented no longer $a$ and $a'$, instead $a_0$ and $a_1$, avoiding ambiguity with the use of same notation for the sample action of $\pi(\cdot|s)$.
>
> 3. More details are added to show how the two-action case could extend to the infinite-action case.
>
> > **4. It is necessary to elaborate more on why OVD-Explorer has potential to deal with the over-exploration issue.**
>
> We’d like to clarify that mutual information to optimal value function[3] (or action[4]) is widely used to design exploration policy and achieve relatively good performance. The potential for OVD-explorer to tackle with the over-exploration issue is attributed to approximating the optimal value function by formulating epistemic and aleatoric uncertainty explicitly. Further, the detailed analysis in Sec. 3.3 and experiments in Sec. 4 show that OVD-explorer avoids the low information gain areas with relatively high aleatoric uncertainty and low epistemic uncertainty.
>
> > **5. It would be great that some theoretical guarantee can be showed for OVD-Explorer.**
>
> We would like to clarify that in our key theoretical part Theorem 1,  a general and practically effective approach [5] is used to approximate $p(a|\bar{z}(s,a),s)$. It is indeed not based on rigorous mathematical argument, but has been widely used in [6, 7, 8] and achieves relatively good performance. Moreover, we follow the intuition that "not only select the action that leads to areas with high uncertainty, but also avoid the ones that only lead to the area with high aleatoric uncertainty".
> Intuitively, using the approach in [5], we could regularize the target policy with the upper bound estimations of the return distribution, and the behavior policy could explore more informative state-action regions.
>
> Hence, we believe that our method is practically effective and intuitively reasonable.
>
> We hope we have clarified your concerns. Looking forward to further discussions with you.
>
> ---
>
> [1] Nikolay Nikolov, Johannes Kirschner, Felix Berkenkamp, Andreas Krause, Information-Directed Exploration for Deep Reinforcement Learning, ICLR 2019.
>
> [2] Xiuyuan Lu, Benjamin Van Roy, Vikranth Dwaracherla, Morteza Ibrahimi, Ian Osband, Zheng Wen, Reinforcement Learning, Bit by Bit, 2021.
>
> [3] Wang, Zi, and Stefanie Jegelka. "Max-value entropy search for efficient Bayesian optimization." ICML. PMLR, 2017.
>
> [4] Russo D, Van Roy B. Learning to optimize via information-directed sampling. Advances in NIPS, 2014, 27: 1583-1591.
>
> [5] Wang, Zi, and Stefanie Jegelka. "Max-value entropy search for efficient Bayesian optimization." ICML. PMLR, 2017.
>
> [6] Syrine Belakaria, Aryan Deshwal, and Janardhan Rao Doppa. Max-value entropy search for multi-objective bayesian optimization with constraints.CoRR, abs/2009.01721, 2020
>
> [7] Perrone, Valerio, et al. "Constrained Bayesian optimization with max-value entropy search." arXiv preprint arXiv:1910.07003 (2019).
>
> [8] Li, Shibo, et al. "Multi-fidelity Bayesian optimization via deep neural networks." Advances in NIPS 33 (2020).

---

### Author Response · Authors · 2021-11-19
**Looking forward to discussing with the reviewers further.**

Dear Reviewers,

Thanks for your careful and valuable comments that we've used to greatly improve the paper. We have responded to all the concerns that you have mentioned, and submitted a revision in which the main modifications are colored in red.

Reviewer nxh5 and Reviewer Sfg5 pointed out the ambiguity about notation $a$ in the proof section. We are very sorry about this and have very carefully clarified the notations in the proof to make it unambiguous and easy to read. The main concern Reviewer xZHr raised is about intuition, and we have clarified this from several perspectives, including discussion with Hoeffding-based exploration and Bernstein-based exploration, experiment results, and the other reference papers. Per Reviewer YS7S's suggestion, we conducted the experiments in the DeepMind Control suite domain.

Please see our detailed responses below and we hope we have addressed all your concerns. We would greatly look forward to further discussing with you.

Regards
Authors

---

### Decision · Program_Chairs · 2022-01-20

**Decision:**

Reject

**Comment:**

The reviewers found the work interesting but have concerns about the correctness of some of the claims in the paper. Also some reviewers would like to see more experiments and some have concerns about the theoretical results. Overall, I see the work promising but it requires a major revision and some improvements to pass the bar. I would recommend the authors to use the reviewers' comments and prepare the paper for future venues.